# System-wide identification and prioritization of enzyme substrates by thermal analysis

Amir Ata Saei [1,2,11✉], Christian M. Beusch [1,11], Pierre Sabatier [1], Juan Astorga Wells [1], Hassan Gharibi [1], Zhaowei Meng[1], Alexey Chernobrovkin [1,3], Sergey Rodin[1,4], Katja Näreoja[5], Ann-Gerd Thorsell[5], Tobias Karlberg[5], Qing Cheng[6], Susanna L. Lundström[1], Massimiliano Gaetani [1,7,8], Ákos Végvári [1,9], Elias S. J. Arnér [6], Herwig Schüler [5] & Roman A. Zubarev [1,10✉]

Despite the immense importance of enzyme–substrate reactions, there is a lack of general and unbiased tools for identifying and prioritizing substrate proteins that are modified by the enzyme on the structural level. Here we describe a high-throughput unbiased proteomics method called System-wide Identification and prioritization of Enzyme Substrates by Thermal Analysis (SIESTA). The approach assumes that the enzymatic post-translational modification of substrate proteins is likely to change their thermal stability. In our proof-of-concept studies, SIESTA successfully identifies several known and novel substrate candidates for selenoprotein thioredoxin reductase 1, protein kinase B (AKT1) and poly-(ADP-ribose) polymerase-10 systems. Wider application of SIESTA can enhance our understanding of the role of enzymes in homeostasis and disease, opening opportunities to investigate the effect of post-translational modifications on signal transduction and facilitate drug discovery.

[1] Division of Physiological Chemistry I, Department of Medical Biochemistry and Biophysics, Karolinska Institutet, Stockholm, Sweden. [2] Department of Cell Biology, Harvard Medical School, Boston, MA, USA. [3] Pelago Bioscience AB, Solna, Sweden. [4] Department of Surgical Sciences, Uppsala University, Uppsala, Sweden. [5] Department of Biosciences and Nutrition, Karolinska Institutet, Huddinge, Sweden. [6] Division of Biochemistry, Department of Medical Biochemistry and Biophysics, Karolinska Institutet, Stockholm, Sweden. [7] SciLifeLab, Stockholm, Sweden. [8] Chemical Proteomics Core Facility, Division of Physiological Chemistry I, Department of Medical Biochemistry and Biophysics, Karolinska Institutet, Stockholm, Sweden. [9] Proteomics Biomedicum, Division of Physiological Chemistry I, Department of Medical Biochemistry and Biophysics, Karolinska Institutet, Stockholm, Sweden. [10] Department of Pharmacological & Technological Chemistry, I.M. Sechenov First Moscow State Medical University, Moscow, Russia. [11]These authors contributed equally: Amir Ata Saei, Christian M. Beusch. ✉email: Amirata.Saei.Dibavar@ki.se; Roman.Zubarev@ki.se

At least a third of all proteins possess enzymatic activity. One of the most comprehensive enzyme databases BRENDA comprises >9 million protein sequences and encompasses 6953 classes of enzyme-catalyzed reactions[1]. Many of these enzymes catalyze the modifications of protein substrates. Only in human genome, an estimated 1089 non-metabolic enzymes are present[2], including for example more than 500 putative kinases. Transient modulation of protein post-translational modifications (PTMs) controls numerous cellular processes by inducing a host of downstream effects, such as changes in protein function, stability, interactions, hemostasis, localization, and cellular diversification[3]. Not surprisingly, mechanisms and kinetics of protein modifications have become a vibrant research area. However, there is still a lack of the methods for identifying substrates undergoing structural changes due to modifications[4–6]. Characterization of enzyme–substrate associations is essential for our understanding of cell biology and disease mechanisms; moreover, many high-throughput drug screening assays rely upon modified substrates as a readout[7,8]. The deficit of information on the physiological substrates of enzymes hampers the development of effective therapeutics, e.g. in Parkinson's disease[9] and cancer[10]. Sometimes, however, there are too many known substrates, and an intelligent prioritization approach is needed for further exploration.

Most existing techniques used for identifying specific substrates are enzyme-specific, labor-intensive, and often not straightforward. Such approaches include the use of genetic and pharmacologic perturbations[11], substrate-trapping mutants[12], affinity purification-mass spectrometry[13], utilizing peptide[14] or protein arrays[15], tagging the client proteins by substrate analogs using engineered enzymes[16] and peptide immunoprecipitation[17] or the employment of sophisticated computational tools[18]. Most of these techniques are specifically designed for a certain enzyme or enzyme class, which limits their applicability. Engineering enzymes can alter the biology of the system, potentially introducing a bias. Therefore, designing an unbiased, general, and quick proteome-wide method not involving artificial modification of the enzyme or cosubstrate can prove to be a significant methodological advancement and a complement to the above approaches.

Mass spectrometry based CEllular Thermal Shift Assay or Thermal Proteome Profiling (TPP) is a recent method that can assess system-wide protein binding to small molecules, metabolites, or nucleic acids by monitoring changes in protein thermal stability[19,20]. Similar techniques such as stability of proteins from rates of oxidation and limited proteolysis can also be applied to detect metabolite-protein interactions[21,22]. Since PTMs can also alter protein thermal stability, these methods can be potentially used to probe the proteome-wide effects of PTMs. For example, Nordlund et al. have shown that phosphorylation leads to extensive intramolecular reorganization and stabilization of retinoblastoma-associated protein 1 (RB1)[23], while Savitski et al. have shown a correlation between phosphorylation and protein stability in mitosis[24]. By employing CETSA with a western blot readout at a single protein level, it has been shown that O-GlcNAcylation enhances stability of Nod2 protein[25]. Huang et al. have recently developed a method called Hotspot Thermal Profiling that relies on the shifts in peptide melting temperature in response to site-specific phosphorylation sites (hotspots)[5]. In this approach, after the thermal treatment of living cells and isolation of the soluble proteome, the lysate is divided in two aliquots for separate bulk proteome (5%) and phosphopeptide (95%) analyses, to uncover the link between the protein thermal stability and the phosphorylation state of that protein. An assertion made in this work is that the larger the shift, the more likely is the biological importance of a given PTM.

In many cases, the concomitant protein–enzyme and protein–cosubstrate interactions can mask modification-specific thermal stability changes of the substrates. This problem is addressed in our method of System-wide Identification of Enzyme Substrates by Thermal Analysis (SIESTA). SIESTA identifies specific thermal stability changes induced in substrate proteins by a combination of enzyme and cosubstrate as compared to the changes induced by either enzyme or cosubstrate alone (workflow in Fig. 1). The idea of specific response is borrowed from our methods of Functional Identification of Target by Expression Proteomics[26] and ProTargetMiner[27]. Using orthogonal partial least squares-discriminant analysis (OPLS-DA)[28], we create models where the proteins' $Tm$ values for the enzyme + cosubstrate treatment are contrasted against those of the other

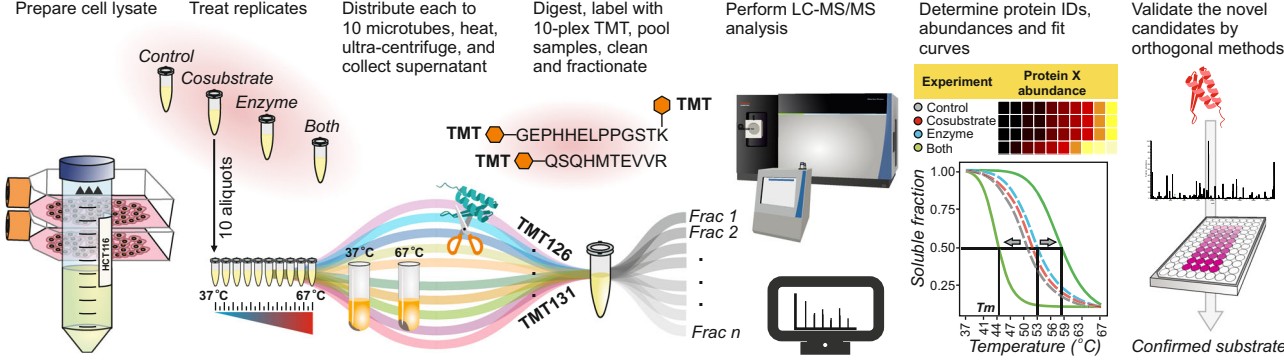

**Fig. 1 SIESTA workflow for unbiased proteome-wide identification and prioritization of enzyme substrates.** A master cell lysate is prepared by multiple freeze-thawing in a non-denaturing buffer. The cell lysate aliquots are treated with vehicle (control), cosubstrate, enzyme, or combination of enzyme with cosubstrate (both). After treatment, each aliquot is split into ten tubes, with each tube heated to a temperature point in the range from 37 to 67 °C. After removing unfolded proteins by ultracentrifugation, identical volumes of supernatants are digested with trypsin. The samples are then serially labeled with 10-plex tandem mass tag (TMT) reagents, pooled, cleaned, and fractionated by reversed-phase chromatography. After LC-MS/MS analyses of each fraction, protein IDs and abundances are determined, and sigmoid curves are fitted through an automated algorithm to determine the melting temperature ($T_m$) for each protein. For each non-vehicle treatment, the read-out is the protein's $\Delta T_m$ shifts (of both signs) compared to control. Any protein shifting more upon addition of enzyme and cosubstrate compared to when they are added alone, are putative substrates of the enzyme under study. Such candidate protein substrates are subsequently confirmed by orthogonal verification methods, starting from the proteins exhibiting largest $\Delta T_m$ shifts and/or involved in relevant pathways.

groups: enzyme-treated and cosubstrate-treated lysates. The proteins on the extremities of such a model will be the potential substrates of the enzymatic system under study. Since OPLS-DA considers both significance and magnitude of an effect at once, selection of substrate candidates this way is statistically robust. We will further elaborate on OPLS-DA below. Here, we apply SIESTA to three distinct enzymes, showing that this method can reveal both known and putative novel substrates that change their stability upon modification in each system and rank them by the probability of having biological impact.

## Results

**SIESTA identified multiple known and putative TXNRD1 substrates**. As a proof of principle, we selected an enzymatic reaction involving an oxidoreductase. For this reaction we employed human selenoprotein thioredoxin reductase 1 (TXNRD1), a key enzyme that catalyzes the reduction of specific substrate proteins using nicotinamide adenine dinucleotide phosphate (NADPH) as a cosubstrate[29]. Since such reduction reaction should destabilize substrate proteins and lead to mostly negative $\Delta T_m$, the asymmetry between positive and negative $\Delta T_m$ values will be easy to verify. A SIESTA experiment was performed in HCT116 cell lysate treated in duplicates with vehicle, NADPH, TXNRD1, or both (Supplementary Data 1).

Changes in $T_m$ after NADPH treatment revealed the shift in the stability of 287 proteins (Fig. 2a), several of which were known NADPH binders. IDH1 is shown as a known example in Fig. 2b. The reproducibility between the replicates is shown in Supplementary Fig 2a. Among the 40 proteins annotated as NADPH binders in UniProt database, 30 proteins (75%) were verified in our experiment which indirectly confirmed the validity of our experiment (Supplementary Fig. 2a). 247 novel proteins were identified as putative NADPH binders (Supplementary Data 1).

The analysis of $\Delta T_m$ shifts in the TXNRD1 + NADPH treatment revealed the destabilization of both known and novel substrate proteins (Supplementary Data 1). In general, as expected, there was an asymmetry in $T_m$ shifts in favor of destabilization (63 destabilized proteins vs. 15 stabilized) (Fig. 2c).

OPLS-DA is a multivariate supervised modeling tool for pinpointing the variables (here proteins) that have the largest discriminatory power between the two or more statistical groups (samples)[28]. In the loading plot or score scatter plot for two-group comparison models, the predictive component is the x-axis, while y-axis is related to the orthogonal components that is irrelevant in this study (Supplementary Fig. 1a). In the loading plot (Supplementary Fig. 1b), each protein is represented by a dot. In SIESTA, the protein $T_m$s for single treatments are contrasted with those from the combination treatment. The blue stars on either side of the plot are the reference points for the treatments. Therefore, proteins specifically stabilized by the modification will move close to the reference point of the combination treatment and the destabilized proteins will be further away on the opposite side. The proximity of a protein to the reference point on either side of the x-axis is a measure of the magnitude of the thermal stability change upon modification and its reproducibility among the replicates. Each protein can also be characterized by the variable influence on projection (VIP-value). The VIP-values quantify the impact each variable (i.e. protein) has on the OPLS-DA model, with a higher value corresponding to a greater contribution[27,30]. Thus the proteins with the highest VIP values are suitable as candidates for validation. For more detailed explanation, see Umetrics documentation[27,31].

An OPLS-DA model contrasting TXNRD1 + NADPH with enzyme and cosubstrate single treatments was also used to reveal the specifically shifting proteins and rank them by their shifts and VIP-values (Fig. 2d, Supplementary Data 1; Supplementary Fig. 2b).

Examples of melting curves for proteins destabilized by TXNRD1 are shown in Fig. 2e. The 78 identified putative substrates mapped to the following INTERPRO Protein Domains and Features pathways: Thioredoxin-like superfamily (11 proteins, $p = 1.3e{-}09$) and Thioredoxin domain (7 proteins, $p = 8.7e{-}08$). The prediction of new substrates for TXNRD1 is not surprising, as the mammalian TXNRD1 is known to have an easily accessible and highly reactive selenocysteine-containing active site[32].

GPX1 was the protein showing the strongest destabilization. Some GPX isoenzymes are known to be directly reduced by TXNRD1[33]. Among the identified TXNRD1 substrates, TXNL1 (or TRP32)[34] and NXN[35] are well known. Note that secondary reactions are unlikely during the SIESTA procedure, as the typical cellular volume is diluted ≈77-fold. That is why cell lysates are used for discovery of direct interactions by thermal profiling without the fear of secondary reactions[36]. Furthermore, if secondary reactions were present, they would also occur in lysates treated with NADPH alone (as the basal levels of cellular TXNRD1 was also present there), and thus would be filtered away in our analysis.

To test our candidate selection procedure and determine the false discovery rate (FDR) among the SIESTA hits, we performed permutation of the protein $Tm$ values within the dataset and applied to thus obtained nonsensical datasets the same selection criteria as for the unshuffled data. On average, 12 new proteins passed the criteria, which gave the FDR of 15%. Note that the top candidates with the largest $\Delta T_m$ values must have a much lower FDR.

To verify that the identified proteins can be directly reduced by TXNRD1, we designed a sequential iodoTMT labeling approach, with which the reduction/oxidation can be quantitatively analyzed on the single cysteine level. For this purpose, we incubated the recombinant candidate proteins GPX1, GPX4, GSTO1, GSTO2, PRDX2, PRDX6, and GULP1 with TXNRD1 + NADPH under the same conditions as in the SIESTA experiment. The results confirmed that GPX1, GPX4, GSTO2, PRDX2, and GULP1 can be directly reduced by TXNRD1 (Fig. 2f, Supplementary Fig. 2c for GPX4 and Supplementary Data 2). For example, in PRDX2 both Cys51 and Cys172, which form an interchain disulfide bond[37], were found reduced. GULP1 was reduced on Cys115 by TXNRD1. Interestingly, GULP1 exists as a dimer in vivo[38] and we noted an increased monomer ratio to total protein for GULP1 upon incubation with TXNRD1 + NADPH (Supplementary Fig. 2d–e). We however could not confirm the reduction of GSTO1 and PRDX6. This might be due to the absence of certain peptides in the MS data. The fact that PRDX2 was detected here as a substrate for TXNRD1 indicates that the enzyme has a capacity to also directly reduce disulfides in this protein to some extent. Indeed, the validation redox proteomics experiment showed that TXNRD1 + NADPH can reduce PRDX2 (Fig. 2f).

There were several proteins which were stabilized in the TXNRD1 + NAPDH treatment, such as CYB5R2 and ACADM. This stabilization might be due to the protein interaction with the reduced form of TXNRD1, or because reduction of some substrates might lead to oxidation of others. Some proteins, such as TXN and TXNDC17, two known substrates of TXNRD1[39], were absent in the SIESTA output due to their deviating melting behavior. For example, although TXN was quantified in all replicates, it remained on average 63% soluble even at 67 °C. Therefore, it was not possible to measure its $T_m$ by fitting a sigmoidal curve, and thus TXN was automatically excluded from the analysis (also TXN also did not melt well in the PARP10 and

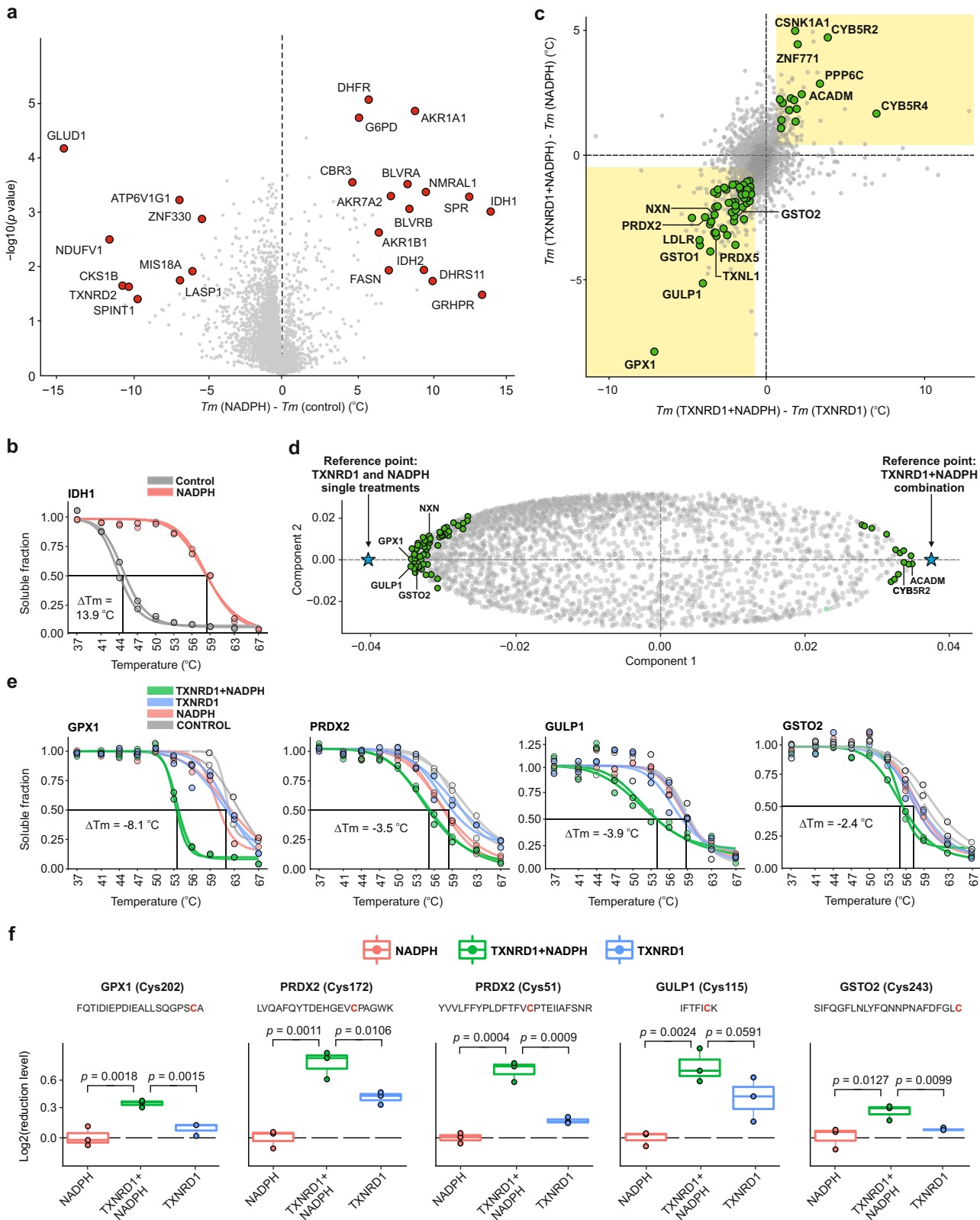

AKT1 experiments in HCT116 and HELA cells). Thioredoxin reductase is known from literature to reduce protein disulfide isomerase (PDI)[40]. We quantified all six PDIAs, of which only PDIA6 was destabilized, but only by −0.69 °C and was therefore excluded by our stringent criteria. Whether these two proteins are substrates of the human TXNRD1 is yet to be seen.

**SIESTA identified novel putative substrates for protein kinase B (AKT1).** In order to confirm the utility of SIESTA for phosphorylation as a ubiquitous and small modification, we chose the AKT1 (protein kinase B) as a model system due to its importance in metabolism, proliferation, cell survival, growth, and angiogenesis. In AKT1 SIESTA experiment (data in Supplementary

**Fig. 2 Proof-of-principle SIESTA experiment revealed known TXNRD1 substrates and suggested novel candidates. a** Scatterplot of protein $T_m$ differences upon addition of NADPH in lysate. Known proteins from UniProt are shown in red ($n = 2$ independent biological replicates; two-sided Student $t$-test; no adjustment for multiple comparisons was performed). **b** Representative stabilization of NADPH binding protein IDH1. **c** Scatterplot of $T_m$ differences reveals the $T_m$ shifts occurring only after simultaneous TXNRD1 + NADPH addition; these shifts are thus likely due to enzymatic modifications (yellow shaded area, known and putative substrates are shown as green circles). **d** Potential substrates (green circles) are mostly located close to the negative reference point (blue star) in an OPLS-DA model contrasting the TXNRD1 + NADPH $T_m$ against the single treatments. Proteins shown in green are those identified as substrates in **c**. **e** Representative melting curves of GPX1, PRDX2, GULP1, and GSTO2 are shown. **f** Reduction of cysteines in the substrate proteins by incubation with TXNRD1 + NADPH ($n = 3$ independent biological replicates, one-sided Student $t$-test), measured by sequential iodoTMT labeling (Center line—median; box limits contain 50% of data; upper and lower quartiles, 75 and 25%; maximum—greatest value excluding outliers; minimum—least value excluding outliers; outliers—more than 1.5 times of the upper and lower quartiles) (NADPH nicotinamide adenine dinucleotide phosphate, $Tm$ melting temperature, TXNRD1 thioredoxin reductase 1). Source data are provided as a Source Data file.

Data 3), ATP was used at 500 μM, at which concentration it only acts as a cosubstrate[41]. Out of the 380 ATP-binding proteins identified in our experiment, 123 were annotated in UniProt as ATP binders and 161 were also identified in a two-dimensional TPP experiment on ATP in Jurkat cells[41]. 257 proteins were identified as novel putative ATP binders (Supplementary Fig. 3a, Supplementary Data 3). The melting curves of known ATP interactors ACTB and MAP2K4 are shown in Supplementary Fig. 3b.

In total, 44 proteins were identified as putative AKT1 substrates (Fig. 3a, Supplementary Data 3), among which TRIP12, MEF2D, COPS6, and BCL3 proteins were known. Interestingly, BCL3 is known to be specifically modified and stabilized by AKT1[42].

The melting curves for representative substrates are shown in Fig. 3b. Among the remaining 40 proteins, 4 molecules were known to interact with AKT1 (CAMKK1, PLEKHF2, CEP76, and IMPDH2). Note that in alternative methods, such as KISS[43], any protein interacting with a kinase is considered to be a potential substrate[44]. OPLS-DA loading rankings of the putative substrates and their VIP-values are provided in Supplementary Data 3 (Supplementary Fig. 3c).

Since we used here a universal 1 °C $\Delta T_m$ minimum cutoff for selection of substrates, setting a smaller modification-specific cutoff (e.g., for phosphorylation as a small modification), would increase the number of identified substrate candidates, at the expense of somewhat higher false positive rate. For example, reducing the cutoff to 0.5 °C increases the number of AKT1 substrate candidates by 28, adding 5 more known substrates (TBC1D4, PLK1, PDE3A, SSBP4, and AKT1 itself). It should also be considered that phosphorylation is a dynamic modification, and the experimental conditions and the choice of cell lines affects the list of substrate candidates. Furthermore, not all proteins or phosphopeptides are detected in all experiments. Therefore, the lists of AKT1 substrates revealed in different studies sometimes show few overlaps. For example, in two direct microarray screenings of human cells, 165[45] and 51[46] AKT1 substrates have been found, respectively, with no overlap between them. In the PhosphoSitePlus database[47] containing an accumulated list of 206 human protein substrates for AKT1, there are 4 and 7 proteins, respectively, that are overlapping with the two mentioned studies, while the 72 SIESTA substrates obtained with a 0.5 °C cutoff, similarly gave 4 overlaps.

In should be noted that recent findings from two independent groups[48,49] raised doubts in the extent of stability-altering phosphorylation postulated by Huang et al.[5], indicating that the percentage of phosphorylation events leading to shift in protein stability is lower than expected.

To validate a number of selected putative substrates, we incubated recombinant PLEKHF2 and TRAPPC2L with AKT1 + ATP under the exact conditions as in the SIESTA experiment. PLEKHF2 is already known to interact with AKT1[50]. Tracking

phosphate release from these recombinant proteins confirmed their modification by AKT1 (Supplementary Fig. 3d).

To further validate the phosphorylation events in living cells, we treated HELA cells with vehicle or two AKT1 inhibitors (AKT1/2 inhibitor (1,3-Dihydro-1-(1-((4-(6-phenyl-1H-imidazo[4,5-g]quinoxalin-7-yl)phenyl)methyl)-4-piperidinyl)-2H-benzimidazol-2-one trifluoroacetate salt hydrate) and Ipatasertib) for 2 h in a concentration series and analyzed the phosphopeptides (Supplementary Data 4). As shown in Fig. 3c, there is an obvious reduction of phosphorylation levels in known and putative substrates of AKT1 identified in SIESTA with increased concentration of both compounds. We found peptides with significantly altered phosphorylation levels from at least six proteins detected in SIESTA as AKT1 substrates. These six proteins are BCL3, MEF2D, TRIP12, MACF1, IREB2, and LNPEP; of these, BCL3, MEF2D, and TRIP12 are known substrates of AKT1 as indicated above. The altered phosphopeptides as well as a cumulative analysis of other phosphopeptides from the same protein are shown for the four proteins in Fig. 3c. Our data show the modification of TRIP12 on several sites, with phosphorylation on Ser77 and Ser85 being quantified multiple times. These data validate that the identified hits in SIESTA are actual cellular substrates of AKT1.

**SIESTA identified and ranked many novel putative substrates for PARP10.** We next selected the poly-(ADP-ribose) polymerase-10 (PARP10) system that performs mono-ADP ribosylation of proteins[51]. ADP-ribosylation is involved in cell signaling, DNA repair, gene regulation, and apoptosis. Identification of PARP family substrates by mass spectrometry has generally proved challenging, as ADP-ribosylation is a glycosidic modification that can be easily lost during protein extraction or sample processing. It is also highly labile in the gas phase, which hampers its detection by MS/MS. Different strategies have thus been used to enrich the modified peptides for mass spectrometric analysis and to employ gentle MS/MS methods[52,53]. However, since ADP-ribosylation is a rather large modification, it should be amenable to SIESTA.

Among the proteins shifting with the cosubstrate nicotinamide adenine dinucleotide (NAD) (data in Supplementary Data 5), 22% (9/41) of the proteins annotated as NAD binders in UniProt were found, together with 87 putative new NAD binding proteins (Supplementary Fig. 4a, listed in Supplementary Data 5). Data on CTBP2 and GALE are shown as known examples in Supplementary Fig. 4b.

In total, 58 proteins were identified as potential PARP10 substrates (Fig. 4a, Supplementary Data 5). Some of these candidates were already known, such as ILF2, ILF3, IPO4, and PUM1[54] as well as GAPDH[55]. Melting curves for some of the putative substrates are shown in Fig. 4b. An OPLS-DA model contrasting PARP10 + NAD $T_m$ vs. those from all other

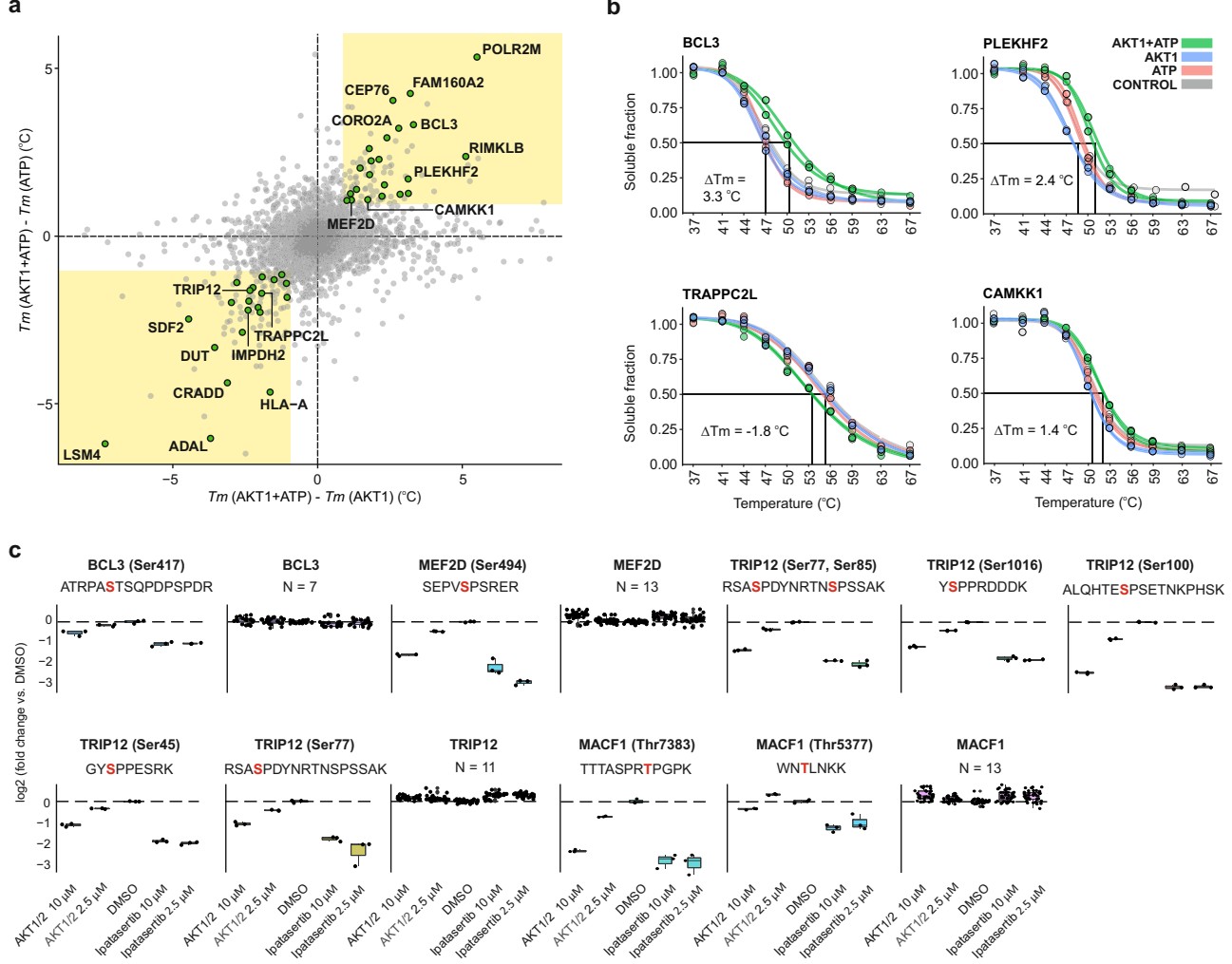

**Fig. 3 SIESTA identified known and putative substrates for AKT1 kinase. a** Scatterplot of $T_m$ differences reveals the $T_m$ shifts occurring only after simultaneous AKT1 + ATP addition (known and putative substrates are shown as green circles). **b** Representative melting curves for known and putative AKT1 substrates. **c** The reduction in the phosphorylation level of AKT1 substrate proteins by AKT1/2 inhibitor and ipatasertib treatment at 2.5 and 10 μM (all shown peptides are significant under all compound treatment conditions vs. DMSO condition ($p < 0.05$); $n = 3$ independent biological replicates, two-sided Student $t$-test). The phosphorylation levels of other phosphopeptides detected for the same proteins are compiled into one plot. $N$ denotes the number of unchanged phosphopeptides for each protein. (Center line—median; box limits contain 50% of data; upper and lower quartiles, 75 and 25%; maximum—greatest value excluding outliers; minimum—least value excluding outliers; outliers—more than 1.5 times of the upper and lower quartiles) (AKT1 RAC-alpha serine/threonine-protein kinase or protein kinase B, ATP adenosine triphosphate, $T_m$ melting temperature). Source data are provided as a Source Data file.

treatments is given in Supplementary Fig. 4c, and the OPLS-DA loading rankings of the putative substrates and their VIP values can be found in Supplementary Data 5.

The majority of the identified PARP10 substrates were novel, reflecting the limited number of previous studies in this area. We used mass spectrometry to verify the PARP10-mediated mono-ADP-ribosylation of destabilized PDRG1 and HDAC2 as well as the stabilized PIN4 and CASP6, selected based on the OPLS-DA rankings and availability of full-length recombinant proteins. We also validated RFK, as it showed a dramatic stabilization when compared to NAD and weak stabilization when compared to the PARP10 treatments. After incubation with recombinant PARP10 + NAD, the above proteins were digested and analyzed with LC-MS/MS. Every higher energy collision dissociation MS/MS event triggered in data-dependent acquisition was investigated in real time for the presence of signature ions of adenine ($m/z$ 136.0623), adenosine-18 ($m/z$ 250.094) and adenosine monophosphate

(AMP, $m/z$ 348.0709). The presence of these ions would then trigger a second MS/MS event using electron-transfer dissociation (ETD) with a supplementary HCD activation (EThcD). The obtained PDRG1 sequence coverage was 74%, and the protein was found modified with ADP-ribose in three locations: on Glu110, Glu75, and Asp32 (Supplementary Table 1). The ETD MS/MS spectrum of a peptide with Glu110 is shown in Fig. 4c and the other sites are shown in Supplementary Fig. 5a–b. The RFK sequence coverage was 94%, and ADP-ribose moieties were found in three positions: on Glu140, Glu131, and Glu113, ordered from the highest to the lowest peptide score (Supplementary Table 1). The ETD MS/MS spectrum of the peptide with highest score is shown in Supplementary Fig. 4d and the other sites are shown in Supplementary Fig. 5c–d. For HDAC2 and PIN4, the sequence coverage with trypsin digestion was not sufficient to observe all potential modifications. Consequently, the ADP-ribosylation of HDAC2, PIN4 as well as PDRG1 and RFK was

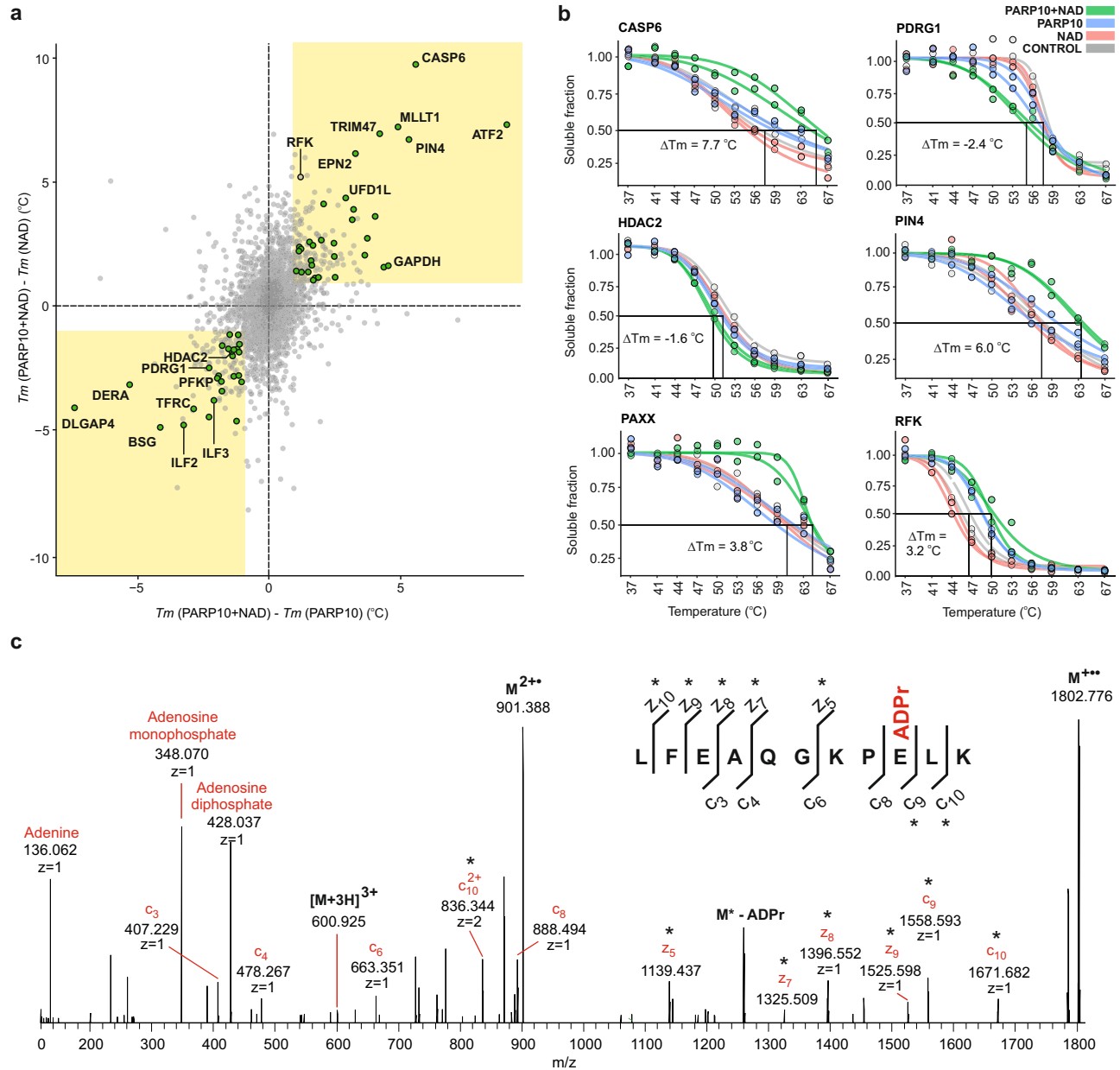

**Fig. 4 SIESTA identified and ranked known and novel PARP10 substrates. a** Scatterplot of $T_m$ differences reveals the shifts occurring when PARP10 + NAD are incubated with cell lysate. Known and putative substrates are shown in green circles. **b** Representative melting curves of putative PARP10 substrates. **c** Mono-ADP-ribosylation on a glutamic acid residue Glu110 in the PDRG1 peptide with the highest sequence-fitting score revealed by targeted ETD MS/MS of 3+ molecular ions (M). The fragments carrying the modification are marked with an asterisk ($T_m$ melting temperature, NAD nicotinamide adenine dinucleotide, PARP10 poly-(ADP-ribose) polymerase-10). Source data are provided as a Source Data file.

verified using a chemiluminescence assay (Supplementary Fig. 4e).

CASP6 (caspase-6) showed the strongest specific stabilization ($\Delta T_m$ of 7.7 °C, Fig. 4a, b), but its modification was not verified in either of the two in vitro assays. It should be noted that PARP10 was suggested to be a substrate for caspase-6 during apoptosis[56]. PARP10 has a major cleavage site at Asp406 that is preferentially recognized by caspase-6[56]. The strong specific thermal stabilization might therefore indicate that PARP10 induces a conformational change in caspase-6 and thus an increase in its stability by binding, as has been reported for other caspase-6 substrates[57]. The reason why caspase-6 stabilization was not observed upon PARP10 addition in the absence of NAD is that auto-modified PARP10 is required for effective caspase-6 binding[56].

**SIESTA uncovers protein–protein interactions**. A cursory look at the vertical axes on the 2D plots in Figs. 2c, 3a, and 4a shows the presence of proteins that exhibit a shift in stability when only the protein (enzyme) is added to the lysate. Our analysis showed 49, 38, and 182 proteins with significant stability shifts in response to TXNRD1, AKT1, and PARP10, respectively (Fig. 5a, Supplementary Data 6). Representative proteins are shown in Fig. 5b. For the AKT1 system, PHLPP2[58], GET4[59], MAZ[60], and FANCI[61] are known interactors according to the literature. Of these, MAZ is even known to be phosphorylated by AKT1[60]. To further validate if TPP can be used for discovering protein–protein interactions in cell lysate, we performed a pull-down experiment with the His-tagged PARP10 added to the HCT116 cell lysate (Supplementary Data 7). Six (out of 115)

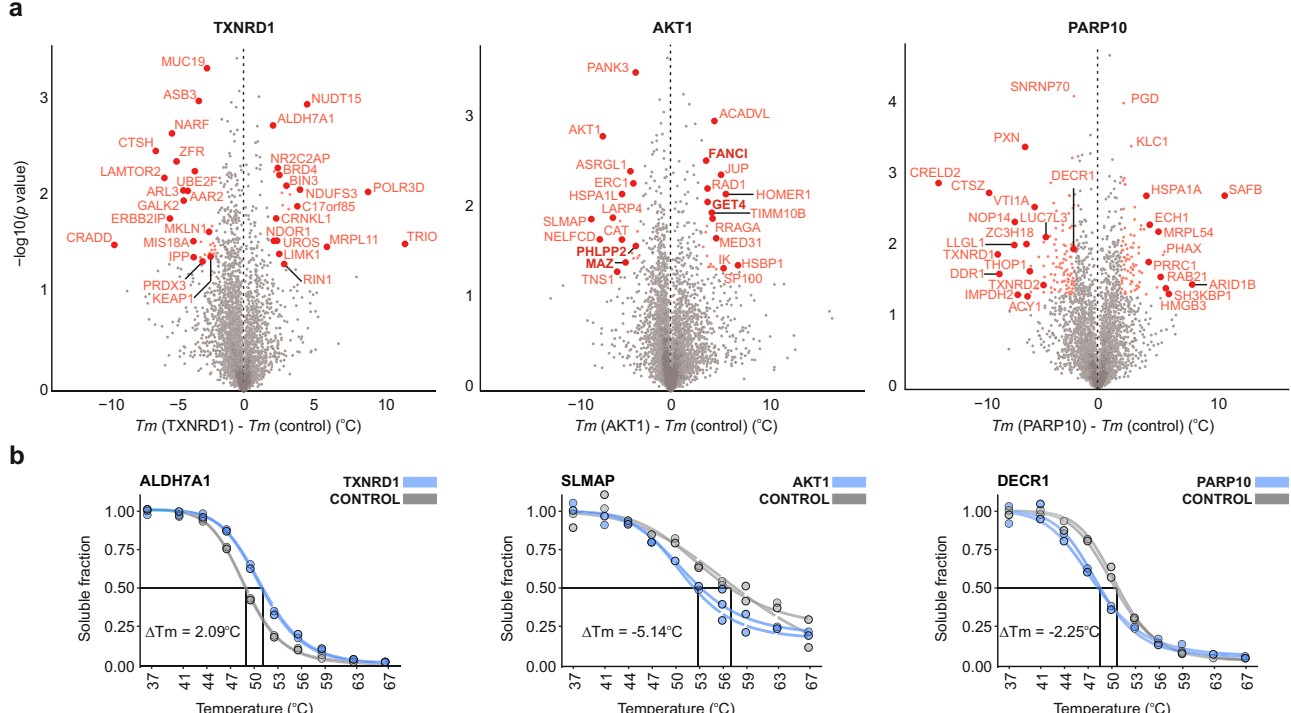

**Fig. 5 TPP identifies protein interaction partners of enzymes added to cell lysates. a** Proteins' $T_m$ shifting with addition of only enzyme to cell lysate. The known proteins interacting with AKT1 are shown in bold red (FUNCI, GET4, PHLPP2, and MAZ) ($n = 2$ independent biological replicates; two-sided Student $t$-test; no adjustment for multiple comparisons was performed). **b** Representative shifting proteins with each enzyme. DECR1 was also enriched in the affinity pulldown experiment with PARP10 ($T_m$ melting temperature). Source data are provided as a Source Data file.

proteins enriched in pulldown were in common with SIESTA data (Supplementary Fig. S6). Since most of the SIESTA-identified interacting proteins have not been mentioned in literature, we conclude that TPP can probe even weak or transient protein–protein interactions that are hard to identify by classical methods, such as pulldowns.

## Discussion
We demonstrate SIESTA to be a general approach for unbiased identification and prioritization of functional protein substrates for specific enzymes in a proteome-wide manner. We uncover several known or novel substrate candidates for TXNRD1, AKT1, and PARP10 enzymes, implicating them in important cellular processes. By applying a ranking system for the putative substrates based on $\Delta T_m$ and OPLS-DA parameters, SIESTA can identify the most plausible substrates for functional validation, which can help systematic uncovering of the biophysical consequences of PTMs. Besides the use in fundamental research, SIESTA can also facilitate drug development by discovering kinetically and energetically favorable substrates in screening for enzyme inhibitors[7,8].

Here we show the applicability of SIESTA for three distinct enzymes, and the utility of SIESTA for other enzyme systems will have to be established in further studies. It can be hypothesized that any modification will have some effect on protein stability, even though some of these changes may be too minute to be easily detectable. Such substrates can be potentially discovered via improving the statistical power of the SIESTA analysis, e.g., by adding more biological replicates. One area where SIESTA might prove particularly useful is in studying PTMs that are difficult to enrich. While SIESTA will discover only substrates that significantly change their thermal stability upon modification, in our

experience such substrates are more likely to be biologically relevant. Paradoxically, this feature makes it difficult to compare SIESTA results with those obtained by other methods that typically lack such ranking ability. Unlike the HotSpot approach that compares the shifts of individual modified peptides with those of the whole bulk protein, thus identifying modifications that may or may not have significant occupancies, SIESTA compares the shift of the whole bulk protein with and without the enzyme and the cosubstrate, thus requiring the majority of protein copy numbers to be modified to be identified as substrate. Therefore, as our results demonstrate, highly ranked SIESTA hits are less likely to fail in subsequent functional validation.

The spatial resolution of the method can be increased by subcellular fractionation of the lysate prior to analysis. Such a strategy could probably identify organelle specific substrates; however, for that the lysis and fractionation should be mild enough to keep the organelle proteins intact. Furthermore, cell- or tissue-specific substrates should be possible to discover by comparing lysates from different sources. Since the addition of enzyme in excess can cause nonphysiological modifications, the identified candidates should be validated by other techniques, as suggested in Fig. 1. Within the current study, for the TXNRD1, AKT1, and PARP10 systems, the ratios of the added enzyme to the same enzyme in the untreated lysate were ~10, ~20, and ~1.5, respectively (Supplementary Fig. S7). Given the subsequent ~77-fold dilution of the lysate, the enzymes were well within their physiological concentrations, which reduced the risk of unspecific reactions. Mild detergents such as NP40 can be used to increase the representation of membrane proteins in SIESTA[62]. On the other hand, the number of missing values can be reduced by using our high-throughput approach to thermal profiling, Proteome-wide Integral Solubility Alteration assay[63]. Affinity purification approaches have the advantage of enriching for low

abundant and low stoichiometry substrates. However, such methods often miss weak binders. This gap can be filled by SIESTA.

One limitation of SIESTA is that the active (or equally activated in each experiment) recombinant enzyme must be used, and it should preferentially act as a monomer or homomultimer. Also, the cosubstrate(s) need(s) to be known and available. On the other hand, SIESTA experimental design can be generalized to accommodate multiple (n) cofactors by introducing control experiments with (n − 1) cofactors and the enzyme. Furthermore, using lysate might distort the spatial regulation of enzyme–substrate interaction and yield substrates that are not active in the biological context. The excess of the enzyme may also lead to unspecific reactions. Moreover, recombinant proteins expressed in Escherichia coli may lack some important PTMs necessary for their activity in human cells.

The studies focused on investigating the effect of PTMs on protein stability are scarce[64,65] and mostly performed on a single protein level. How minuscule changes in chemical structure can lead to conformational and stability changes and manifest signaling and phenotypic consequences, is not fully explored. For example, it has been shown that glycosylation sites on CUB1 domain of Bone Morphogenetic Protein-1 are important for thermal stability and secretion of this protein[64]. Even small modifications such as deamidation can change protein stability against temperature and urea[66]. Therefore, one of the main applications of SIESTA will be to study such events and decipher the biological roles of such modifications in a proteome-wide manner.

The average absolute shifts in substrates for TXNRD1, AKT1, and PARP10 were 2.37, 2.46, and 2.78 °C, respectively, while the median absolute shifts were 2.04, 2.10, and 2.19 °C, respectively. Given the complexity of the PTM cross-talk and the PTM code, the lack of an obvious general trend in $\Delta T_m$ shifts is not surprising, as the entirety of PTMs rather than individual PTM type dictate the structure and energetics of a given protein. This might also explain why we could not verify some known substrates, as it could be simply due to the heterogeneity of the PTM pool. Generalization of the effect of PTMs on protein folding and stability can be difficult, since specific protein-PTM contacts do not necessarily follow general rules and might have evolved to confer beneficial energetic effects on protein folding[65]. For example, N-glycosylation is known to generally increase the stability of target substrates[67], but random introduction of N-glycans in a protein does not necessarily stabilize the protein significantly. The interpretation of disulfide bond reduction on protein stability is more straightforward. However, the impact of other PTMs such as ADP-ribosylation and phosphorylation on protein stability might be harder to understand. Conformational distortion by addition of modification[68], conformational entropy and free energy[69] and size and position of modification[70] as well as changes in the charge state and solvent accessibility determine the outcome of a modification on overall protein stability[70]. Furthermore, multiple modifications might have cumulative effects on the stability of a protein[71] and the final stability of a resultant proteoform is governed by its detailed energetics[72].

Especially in the case of oxidoreductases, SIESTA may prove very useful. Although redox proteomics can also be used to analyze oxidoreductase substrates, as we showed here, it is impossible to cover all cysteine sites in all detected proteins. In contrast, the SIESTA information is obtained at the whole protein level, which by definition encompasses the whole sequence. This said, redox proteomics could still be an effective tool in verification of the oxidoreductase substrates[73].

Finally, we show that SIESTA and thermal profiling in general can be used to discover protein–protein interactions that exert stability changes (in lysate), and that the identified interactions are highly orthogonal to other techniques, such as affinity pulldowns.

Summarizing, the ease, breadth, and speed of identifying enzyme-specific substrates offered by SIESTA can enhance our understanding of enzyme systems and disease, accelerate constructing high-throughput assays and thus facilitate drug discovery. Probing the induced stability changes in substrate proteins, SIESTA workflow provides three layers of information within a single experiment: on cosubstrate-binding proteins, protein–protein interactions, and enzyme substrates.

## Methods

**Cell culture**. Human colorectal carcinoma HCT116 (ATCC, USA) cells were grown at 37 °C in 5% $CO_2$ using McCoy's 5 A modified medium (Sigma-Aldrich, USA) supplemented with 10% FBS superior (Biochrom, Berlin, Germany), 2 mM L-glutamine (Lonza, Wakersville, MD, USA) and 100 units/mL penicillin/streptomycin (Gibco, Invitrogen). Human HELA cells (ATCC, USA) were grown under the exact same conditions in DMEM. Low-number passages were used for the experiments.

**Recombinant proteins**. Human TXNRD1, GPX1, and GPX4 were expressed recombinantly in E. coli and purified as described earlier[74]. PARP10 full-length protein (used in SIESTA) and catalytic domain construct (used in validation assays) were produced as detailed before[75]. The rest of the recombinant proteins were purchased and are listed in Supplementary Table 2 and their identity was validated in the respective mass spectrometry data files.

**SIESTA experiment**. A step-by-step protocol describing the SIESTA experiment protocol can be found at Protocol Exchange[76]. Cells were cultured in 175 cm² flasks, and were then detached, washed twice with PBS, resuspended in 50 mM HEPES pH 7.5, 2 mM EDTA (for TXNRD1) or in 50 mM HEPES pH 7.5, 100 mM NaCl and 4 mM $MgCl_2$ (for PARP10), both with complete protease inhibitor cocktail (Roche). For AKT1 experiment, phosphatase inhibitors (PhosSTOP, Sigma) were also added. Cells were lysed by five freeze-thaw cycles. The cell lysates were centrifuged at 21,000 g for 20 min and the soluble fraction was collected. The protein concentration in the lysate was measured using Pierce BCA assay (Thermo) and the lysate was equally distributed into eight aliquots (1 mL each). For TXNRD1, each pair of samples were incubated with vehicle, 1 mM NADPH, 1 μM TXNRD1, or with TXNRD1 + NADPH at 37 °C for 30 min. For PARP10, each pair of samples were incubated with vehicle, 100 μM NAD, 400 nM PARP10, or with PARP10 + NAD at 37 °C for 1 h. For AKT1, each pair of samples were incubated with vehicle (AKT1 buffer), 500 μM ATP (CAT# GE27-2056-01, sigma), 500 nM AKT1 or with AKT1 + ATP at 37 °C for 30 min. Each replicate was then aliquoted into 10 PCR microtubes and incubated for 3 min in SimpliAmp Thermal Cycler (Thermo) at temperature points of 37, 41, 44, 47, 50, 53, 56, 59, 63, and 67 °C. Samples were cooled for 3 min at room temperature and afterwards kept on ice. Samples were then transferred into polycarbonate thickwall tubes and centrifuged at 100,000 g and 4 °C for 20 min.

The soluble protein fraction was transferred to new Eppendorf tubes. Protein concentration was measured in the samples treated at lowest temperature points (37 and 41 °C) using Pierce BCA Protein Assay Kit (Thermo), the same volume corresponding to 50 μg of protein at lowest temperature points was transferred from each sample to new tubes and urea was added to a final concentration of 4 M. Dithiothreitol (DTT) was added to a final concentration of 10 mM and samples were incubated for 1 h at room temperature. Subsequently, iodoacetamide (IAA) was added to a final concentration of 50 mM and samples were incubated at room temperature for 1 h in the dark. The reaction was quenched by adding an additional 10 mM of DTT. Proteins were precipitated using methanol/chloroform. The dry protein pellet was dissolved in 8 M urea, 20 mM EPPS (pH = 8.5), and diluted to 4 M urea. LysC was added at a 1:100 w/w ratio at room temperature overnight. Samples were diluted with 20 mM EPPS to the final urea concentration of 1 M, and trypsin was added at a 1:100 w/w ratio, followed by incubation for 6 h at room temperature. Acetonitrile (ACN) was added to a final concentration of 20% and TMT reagents were added 4× by weight (200 μg) to each sample, followed by incubation for 2 h at room temperature. The reaction was quenched by addition of 0.5% hydroxylamine. Samples were combined, acidified by TFA, cleaned using Sep-Pak cartridges (Waters), and dried using DNA 120 SpeedVac Concentrator (Thermo). The SIESTA samples for TXNRD1 and PARP10 were then resuspended in 0.1% TFA and fractionated into eight fractions using Pierce High pH Reversed-Phase Peptide Fractionation Kit (Thermo). The AKT1 samples were resuspended in 20 mM ammonium hydroxide and separated into 96 fractions on an XBrigde BEH C18 2.1 × 150 mm column (Waters; Cat#186003023), using a Dionex Ultimate 3000 2DLC system (Thermo Scientific) over a 48 min gradient of 1–63% B (B = 20 mM ammonium hydroxide in ACN) in three steps (1–23.5% B in 42 min, 23.5–54% B in 4 min and then 54–63%B in 2 min) at 200 μL min⁻¹ flow.

Fractions were then concatenated into 24 samples in sequential order (e.g. 1, 25, 49, 73).

**Sequential iodoTMT labeling**. Redox proteomics was adapted from our previous protocol with slight modifications[73]. The proteins (2 μg each, in triplicates) were incubated with 1 mM NADPH, 1 μM TXNRD1, or with TXNRD1 + NADPH at 37 °C for 30 min. After solubilization in methanol, 4.4 mmol L$^{-1}$ of iodoTMT was added to the samples (labels 126, 127, and 128 to replicate 1, 2, and 3 in each treatment) and incubated for 1 h at 37 °C with vortexing in the dark (free SH and SSH groups will be blocked in this stage). The proteins were precipitated using methanol chloroform and after drying, samples were dissolved in Tris buffer with 1% SDS and incubated at 37 °C in the dark with 1 mM DTT for 1 h. Subsequently, the samples were incubated with 4.4 mmol L$^{-1}$ of the second iodoTMT label at 37 °C in the dark for 1 h (labels 129, 130, and 131 to replicates 1, 2, and 3 in each treatment). The reaction was quenched by 20 mM final concentration of DTT. NADPH, TXNRD1, and TXNRD1 + NADPH-treated samples were then individually pooled and precipitated. Protein pellets were dissolved in Tris and urea 8 M. The samples were then diluted to 4 M urea, and lysC was added at a ratio of 1:100 enzyme: protein overnight. After dilution of urea to 1 M, trypsin was added at a ratio of 1:100, followed by incubation for 6 h at 37 °C. Samples were acidified by TFA and cleaned using SepPak and lyophilized using a vacuum concentrator. Samples were dissolved in 0.1% FA and 1 μg of each samples was analyzed with a Q Exactive Plus instrument using a 2 h gradient.

**Phosphoproteomics**. Cells were seeded in 25 cm$^2$ flasks and 24 h later treated with AKT1/2 inhibitor and Ipatasertib at 2.5 and 10 μM for 2 h. Cells were then lysed with 50 mM Tris containing 8 M urea and 1% SDS with phosphatase inhibitor cocktail (PhosSTOP, Sigma). The proteins were then digested like above and peptides were cleaned using SepPak and lyophilized using a vacuum concentrator. Phosphopeptides were then enriched using the High-Select Fe-NTA Phospho-peptide Enrichment Kit (Thermo) according to manufacturer's protocol. The enriched peptides were labeled with TMTpro[77] as detailed above for TMT10 labeling, and then pooled, cleaned, and lyophilized. The samples were then resuspended in 0.1% TFA, separated into eight fractions using Pierce High pH Reversed-Phase Peptide Fractionation Kit (Thermo). The fractionation was performed twice.

**LC-MS/MS**. After drying, samples were dissolved in buffer A (0.1% formic acid and 2% ACN in water). The TXNRD1 and PARP10 samples were loaded onto a 50 cm EASY-Spray column (75 μm internal diameter, packed with PepMap C18, 2 μm beads, 100 Å pore size) connected to the EASY-nLC 1000 (Thermo) and eluted with a buffer B (98% ACN, 0.1% FA, 2% H$_2$O) gradient from 5 to 38% of at a flow rate of 250 nL min$^{-1}$. The eluent was ionized by electrospray, with molecular ions entering an Orbitrap Fusion mass spectrometer (Thermo).

The AKT1 samples were loaded with buffer A onto a 50 cm EASY-Spray column connected to a nanoflow Dionex UltiMate 3000 UHPLC system (Thermo) and eluted in an organic solvent gradient increasing from 4 to 26% (B: 98% ACN, 0.1% FA, 2% H$_2$O) at a flow rate of 300 nL min$^{-1}$ over 95 min.

The iodoTMT labeled samples were loaded with buffer A onto a 50 cm EASY-Spray column connected to an EASY-nLC 1000 (Thermo) and eluted with a buffer B (98% ACN, 0.1% FA, 2% H$_2$O) gradient from 4 to 35% of at a flow rate of 300 nL min$^{-1}$ over 120 min.

The phosphoproteomics samples were loaded with buffer A onto a 50 cm EASY-Spray column connected to a nanoflow Dionex UltiMate 3000 UHPLC system (Thermo) and eluted in an organic solvent gradient increasing from 4 to 32% (B: 98% ACN, 0.1% FA, 2% H$_2$O) at a flow rate of 300 nL min$^{-1}$ over a total of 110 min. The MS parameters of all the above-mentioned experiments as well as the number of quantified proteins are summarized in Supplementary Table 3. The total number of proteins and the number of proteins with missing values in different replicates in each SIESTA experiment are shown in Supplementary Fig. 8.

**Data processing**. Thermo Xcalibur 4.0 was used to control and process the LC-MS data. The raw LC-MS data (SIESTA) were analyzed by MaxQuant, version 1.5.6.5 or 1.6.2.3 (TMTpro)[78]. The Andromeda search engine matched MS/MS data against the UniProt complete proteome database (human, version UP000005640_9606, 92957 entries), unless otherwise specified. Trypsin/P was selected as enzyme specificity. No more than two missed cleavages were allowed. A 1% FDR was used as a filter at both protein and peptide levels. For all other parameters, the default settings were used.

In all SIESTA analyses, TMT10-plex was used for peptide quantification. Cysteine carbamidomethylation was set as a fixed modification, while methionine oxidation was selected as a variable modification. In the AKT1 SIESTA experiments, phosphorylation on serine and threonine was selected as variable modification, and used in quantification. For sequential iodoTMT labeling, iodoTMT6-plex on the MS/MS level was used for quantification of peptide/protein abundances. Methionine oxidation was selected as a variable modification and a customized.fasta file with recombinant protein sequences was used.

Phosphopeptide enriched samples were analyzed in the verification experiments with an in-house modified version of MaxQuant version 1.6.2.3 recognizing TMT16-plex (TMTpro) as an isobaric labeling mass tag. Cysteine carbamidomethylation was set as a fixed modification, while methionine oxidation was selected as a variable modification. Phosphorylation on serine and threonine was selected as variable modification. Pulldown samples for PARP were quantified with the label-free quantification (LFQ) algorithm and the 'match between runs' setting was enabled with default settings. After removing all the contaminants, only proteins with at least two peptides were included in the final dataset.

**Network mapping**. For pathway analyses, STRING version 10.5 protein network analysis tool was used with default parameters[79].

**Validation of mono-ADP-ribosylation by targeted tandem mass spectrometry**. Recombinant RFK (5 μg) and PDRG1 (5 μg) were diluted with 50 Mm HEPES (pH = 7.5), 0.5 mM TCEP, 100 mM NaCl, 100 μM NAD, 4 mM MgCl$_2$, and incubated with 400 nM of PARP10 for 1 h. Proteins were reduced with 10 mM DTT for 30 min and alkylated with 50 mM IAA for 30 min in the dark. Afterwards, 1 M urea was added to the samples and LysC (overnight) and Trypsin (6 h) were added sequentially at 1:100 w/w to protein. After acidification, samples were cleaned using StageTips. Samples were dissolved in 0.1% FA and 1 μg of each samples was analyzed with LC-MS using a 1 h gradient.

The chromatographic separation of peptides was achieved using a 50 cm Easy C18 column connected to an Easy1000 LC system (Thermo Fisher Scientific). The peptides were loaded onto the column at a flow rate of 1000 nL min$^{-1}$, and then eluted at 300 nL min$^{-1}$ for 50 min with a linear gradient from 4 to 26% ACN/0.1% formic acid. The eluted peptides were ionized with electrospray ionization and analyzed on an Orbitrap Fusion mass spectrometer. The survey mass spectrum was acquired at the resolution of 120,000 in the m/z range of 300–1750. The first MS/MS event data were obtained with a HCD at 32% excitation for ions isolated in the quadrupole with a $m/z$ width of 1.6 at a resolution of 30,000. Mass trigger filters targeting adenine, adenosine and AMP ions were used to initiate a second MS/MS event using ETD MS/MS with HCD supplementary activation at 30% collision energy and with a 30,000 resolution. Samples treated with NAD but no PARP10 were used as negative controls.

Spectra were converted to Mascot generic format (MGF) using in-house written RAWtoMGF v. 2.1.3. The MGFs files were then searched against the UniProtKB human database (v. 201806), which included 71,434 sequences. Mascot 2.5.1 (Matrix Science) was used for peptide sequence identification. Enzyme specificity was set to trypsin, allowing up to two missed cleavages. C, D, E, K, N, R, and S residues were set as variable ADP-ribose acceptor sites. Carbamidomethylation was set as a fixed modification on C and oxidation as a variable modification on M.

**In vitro mono-ADP-ribosylation assay**. Hexahistidine-tagged protein substrate was immobilized on Ni$^{2+}$-chelating microplates (5-PRIME). Untagged PARP10 catalytic domain was used for substrate protein modification. Mono-ADP-ribosylation was assessed after incubation with 100 μM NAD$^+$ (including 2% biotinylated NAD$^+$, Trevigen) prior to chemiluminescence detection of biotinyl-ADP-ribose in a Clariostar microplate reader (BMG Labtech) as described in detail before[75].

**Phosphoprotein Phosphate Estimation Assay**. The recombinant proteins were incubated with AKT1 and ATP under exact condition of SIESTA experiment. Samples treated with only AKT1 or ATP were used as controls. Phosphate release from the proteins was measured by the Phosphoprotein Phosphate Estimation Assay Kit (Thermo) according the manufacturer instructions. The absorbance was measured at 620 nm using an Epoch Microplate Spectrophotometer (BioTek).

**SDS-PAGE**. GULP1 (3 μg) was incubated with NADPH (1 mM), TXNRD1 (1 μM) or their combination for 30 min at 37 °C in triplicates. After addition of the NuPAGE LDS Sample Buffer (Thermo), the samples were loaded in a NuPAGE Bis-Tris Mini Gel (Thermo) with 10 lanes and separated on a NuPAGE Bis-Tris 4–12% gel in MOPS SDS Running Buffer under nonreducing conditions at 200 V for 60 min using the XCell SureLock system (Thermo). SeeBlue Plus2 Pre-stained Protein Standard (Thermo) was used as a ladder. The gel was then washed and stained with Coomassie blue for 1 h and then destained overnight. The resulting protein bands were captured using Universal Hood II (Bio-Rad) and analyzed using Quantity One 4.6.9.

**Pulldown for PARP10**. PARP10 interacting proteins were enriched using Pierce™ His Protein Interaction Pull-Down Kit (Thermo) according to the manufacturer protocol. 100 μg of His-tagged PARP10 was incubated with 25 μl of settled HisPur™ Cobalt Resin and incubated at 4 °C for at 30 for immobilization. 500 μg of HCT116 cell lysate (with 10 mM imidazole) was then added to the column, which was then incubated for 1 h at 4 °C with intermittent inversion. The column was washed 7 times. The proteins were then reduced on beads with 10 mM DTT for 30 min and alkylated with 50 mM IAA for 30 min in the dark. Afterwards, urea was added to

the samples to 1 M final concentration and LysC (overnight) and Trypsin (6 h) were added sequentially at 1:100 w/w to protein. After acidification, samples were cleaned using StageTips and dried. Samples were dissolved in 0.1% FA and analyzed with LC-MS over 4 h gradient.

**Statistical analysis**. Most of the data analysis was performed using R project versions 3.6–4.0. Curve normalization and fitting was done by an in-house R package (https://github.com/RZlab/SIESTA; https://doi.org/10.5281/zenodo.4423098)[80]. Briefly, after removing the contaminant proteins and those quantified with less than two peptides, protein abundances in temperature points 41–67 °C were normalized to the total proteome melting curve similar to Franken at al.[36]. For each protein in each replicate, a sigmoid curve was fitted using nonlinear least squares method according to the formula:

$$I \sim (1 - Pl)/\left[1 + \exp\left((T - T_m)/bT\right)\right] + Pl,$$

where Pl—high-temperature plateau of the melting curve, $T_m$—melting temperature, b—slope of the curve.

*P* values for the potential cosubstrate-binding proteins were calculated by *t*-test based on the $T_m$ values between vehicle and cosubstrate replicates. For selection of putative cosubstrate-binding proteins, the following criteria were used: (1) R2 > 0.7 between the measurement and the fitted curve, (2) the standard deviation between the replicates was <2.5 °C, (3) *P* value < 0.05, (4) the absolute mean $\Delta Tm$ larger than 1 °C between the cosubstrate- and vehicle-treated samples.

*P* values for the potential substrates were calculated by two methods. In the first approach, *t*-test was made comparing the protein $T_m$ values in Enzyme + Cosubstrate treatment against Enzyme and Cosubstrate single treatments. For selecting the putative substrates, the following criteria were used: (1) R2 > 0.7 between the measurement and the fitted curve, (2) the standard deviation between the replicates was <2.5 °C, (3) *p* values between the Enzyme + Cosubstrate treatment against Enzyme and Cosubstrate treatments <0.05 for one condition and <0.1 for the other; (4) the absolute mean $\Delta T_m$ was larger than 1 °C for both conditions (a similar approach was used for selection of cosubstrate-binding proteins).

Proper correction for multiple hypothesis requires an a priori knowledge of the error distribution, which was not available due to the complexity of our criteria for choosing SIESTA hits. Thus, to test our candidate selection procedure and determine the FDR among the SIESTA hits, in the pilot experiments, we performed permutation of the protein $T_m$ values between the replicates within the dataset and applied to thus obtained nonsensical datasets the same selection criteria as for the unshuffled data.

The 1 °C cutoff was chosen by analysis of the variation in $T_m$ between replicates. In TXNRD1, AKT1, and PARP10 systems, the median $T_m$ variation for different treatments was 0.50, 0.45, and 0.59 °C, respectively. The proteins passing the significance thresholds were ranked by absolute $\Delta T_m$ or VIP values obtained from OPLS-DA analysis.

For the phosphoproteomics results we used quality threshold by filtering for a 'localization probability' >0.75, an Andromeda search engine score >40 as well as an Andromeda 'Delta score' >8. Peptide abundances were normalized to the total abundance in the corresponding TMTpro channel and log2-transformed.

For analysis of the PARP10 pulldown results, proteins LFQ values were normalized by the total abundance in each sample and missing values were replaced by sampling random numbers from a normal distribution: 1.8 standard deviation down shifted from the mean with a width of 0.25. Proteins with more than 4 missing value (out of the total 8 values) were excluded from statistical analyses. A log2 fold change of 0.5 was selected as a lowest-level cutoff.

Multivariate modeling using OPLS-DA was performed using SIMCA 15.0. Protein loading scores were validated using the VIP values at 95% confidence.

Two-sided *t*-test (with equal or unequal variance depending on *F*-test) was applied to calculate *p* values, unless otherwise specified.

**Reporting summary**. Further information on research design is available in the Nature Research Reporting Summary linked to this article.

## Data availability

The authors declare that all data supporting the findings of this study are available within the paper and its supplementary information files. All relevant data are available from the corresponding authors (A.A.S. and R.A.Z.). The mass spectrometry data that support the findings of this study have been deposited in ProteomeXchange Consortium (https://www.ebi.ac.uk/pride/) via the PRIDE partner repository[80] with the dataset identifiers PXD010554 [http://www.ebi.ac.uk/pride/archive/projects/PXD010554] for PARP10 and TXNRD1 SIESTA, PXD014445 [http://www.ebi.ac.uk/pride/archive/projects/PXD014445] for AKT1 SIESTA, PXD021915 [http://www.ebi.ac.uk/pride/archive/projects/PXD021915] for PARP10 pull-down and PXD021916 [http://www.ebi.ac.uk/pride/archive/projects/PXD021916] for the phosphoproteomics experiment. Source data are provided with this paper.

## Code availability

The curve fitting R package is available in GitHub (https://github.com/RZlab/SIESTA) and via Zenodo (https://doi.org/10.5281/zenodo.4423098) with no access restrictions.

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

## Acknowledgements

We are grateful to Marie Ståhlberg and Carina Palmberg for their assistance in different proteomics experiments. The research is funded by grants from the Knut and Alice Wallenberg Foundation (grant KAW 2015.0063) and VINNOVA (grant Oxidocurin) awarded to R.A.Z.; E.A. is supported by grants from Karolinska Institutet, Swedish Cancer Society and Swedish Research Council, and H.S. by Swedish Research Council (grant 2015-4603). K.N. was supported by a stipend from the Wenner-Gren foundation. A.A.S. was supported by the gold level award from the Thermo Scientific Tandem Mass Tag Research Awards, Swedish Research Council (grant 2020-00687) and Karolinska Institutet (grant FS-2020:0007).

## Author contributions

Conceptualization, R.A.Z. and A.A.S.; methodology and experiment design, A.A.S., R.A. Z., C.M.B, J.A.W., H.S., E.A., S.R. and T.K.; project organization, training, resources, and funding acquisition, R.A.Z., A.A.S., H.S., and E.A; SIESTA experiments, A.A.S., P.S. and M.G.; redox experiments, A.A.S. and P.S.; targeted mass spectrometry for ADP-ribose confirmation, A.A.S. and A.V.; phosphoproteomics, A.A.S., H.G., and Z.M.; pulldown

experiments, A.A.S.; data analysis and visualization, A.A.S., C.M.B., H.G., S.L.L. and A.C.; production and testing of enzymes, A.G.T. and Q.C.; in vitro mono-ADP-ribosylation assays, K.N. and H.S.; writing—original draft, A.A.S. and R.A.Z.; writing—review and editing, A.A.S., R.A.Z., H.S., E.A., and J.A.W.

## Funding

## Competing interests
The authors declare no competing interests.
