## [Peer Review File · Nature Communications]

REVIEWER COMMENTS

Reviewer #1 (Remarks to the Author):

In this work, Ata Saei and colleagues present a new approach to detect enzyme substrates based on thermal proteome profiling. The authors apply this approach to 3 enzymes, recovering some of their known substrates and identifying larger numbers of putative substrates. For each of those, the authors validate a small proportion that they prioritize, showing that the approach is generally valid – with a false positive rate ~20-30%, which I would deem acceptable for a proteome-wide method. The study is generally well designed and clearly written. I only have one major concern with the statistics used throughout the manuscript to identify potential substrates. I also have a couple of suggestions that I feel could make the approach more broadly applicable, and make it clearer for the readership what are the major limitations in its current form.

Major comments

1. The authors used a t-test to calculate significance between conditions (together with a series of additional criteria properly described in the methods). However, it is not clear if the p-values from the t-test were corrected for multiple comparison? From browsing their code on github this does not seem to be the case. If not, this needs to be addressed, since the authors are performing thousands of tests for each experiment. As a suggestion, the authors might want to implement previous strategies, such as the ones in PMID:25278616 or PMID:31582558. The latter goes beyond T_m estimates and might increase the proteome space that the authors can probe (e.g., it could include TXN that the authors discuss does not properly melt).
2. Related to the statistics, I do not fully understand why the authors use the PLS model to rank candidates and not simply the ΔT_m (or the absolute ΔT_m , or a proper significance value). From the methods section it is not clear what is the input for the PLS. Further, from for example Fig 2d the reference point for the single treatment is very close to proteins that show large negative shifts. Shouldn't these proteins be quite far from that reference point?
3. In all 3 examples, the authors mention the (known/putative) co-factor binding proteins, but never mention proteins that are (de)stabilized by the addition of the recombinant protein alone. By browsing the data for the TXNRD1 experiment, I found a few proteins that were strongly stabilized or destabilized by the protein alone (e.g., TRIO, CRADD) – I can imagine that the other two experiments have similar cases. Are these known interactors, or can the authors rationalize these in some way? Perhaps SIESTA could also be used to find new protein-protein interactions, making it even more exciting? It could also facilitate the expansion to heteromultimeric enzymes (see next comment), since the experiments could be performed first with enzyme vs control alone, find the partner enzymes and then use those together for the SIESTA experiment. I understand that this might be outside the scope of the current manuscript, but the authors should at least explore/discuss the results from enzyme vs control experiments.

4. Related to the last point, the authors should acknowledge in the discussion a few more drawbacks of SIESTA in its current form. First, it requires the availability of an active recombinant enzyme (i.e., the enzyme should not need to be activated in any way) that acts as a monomer or homomultimer. Second, the enzyme should only require a single co-factor that needs to be known and available.

Minor comments

1. While the criteria for calling hits in each experiment are explained in the methods section, it would be helpful that in each supplementary dataset there was a column included if a protein is a hit or not.

2. Line 148: "The average correlation between the VIP values of top 100-400 SIESTA proteins with their GeneCard scores (zero value was assigned if the protein was not present in the GeneCard list) was 0.18 ($p < 0.001$)." This sentence is unclear, can the authors explain better what was done here?

3. Line 174: "Interestingly, GULP1 exists as a dimer in vivo ³² and we noted the increased monomer levels for this protein upon incubation with TXNRD1+NADPH (Supplementary Fig. 1c-d)." Can this be caused by the slightly higher levels of TXNRD1 in the TXNRD1+NADPH experiment (by eye this seems to be higher in that experiment)?

4. Line 218: "31% (123/396) of the proteins annotated in Uniprot as ATP binders were also verified in our experiment." The authors could also compare their results to previous TPP experiments with ATP, since even in lysate proteins that are stabilized are not necessarily binding ATP directly.

5. In Fig 2-4, when showing melting curves for potential substrates, I think the controls should also be included.

Andre Mateus

Reviewer #2 (Remarks to the Author):

Saei et al. describe an adaptation of the Thermal Proteome Profiling (TPP) methodology, they term SIESTA and which can be used to particularly screen for enzyme substrates. As the authors point out, proteome-wide screening for substrates of a particular enzyme is not necessarily straightforward.

The authors demonstrate the utility and versatility of SIESTA on three model enzymes, namely the thioredoxin reductase TXNRD1, the protein kinase AKT1, and poly-ADP-ribosyltransferase PARP-10 and show that known as well as novel substrates can be identified for these enzymes using a

'differential' approach. This is certainly an interesting adaptation of the TPP strategy that may address an existing need in screening for enzyme substrates, for instance when studying these enzymes as potential drug targets. Unfortunately, the presentation of the manuscript is far from optimal and a number of points listed below have to be addressed.

Major

1. The authors use a lysate (mild conditions) and add recombinant enzyme + substrate to study in vitro substrates of this enzyme. Although their approach undoubtedly generates interesting data, this strategy may be prone to false positive identifications. PTMs are mainly regulated spatially by bringing substrate and enzyme in close proximity (or not). Now, in a lysate this level of regulation is completely gone. I do not necessarily see a problem in using such a strategy, but I definitely miss a clear statement on this approach in order to make the reader aware of potential shortcomings. For instance, protein/substrate interactions may be specific to a certain organelle. I was first also wondering about secondary effects, which the authors exclude based on the fact that the lysate is 77x fold diluted compared to the cell, but that also means that a huge excess of enzyme is added to drive the reaction which can lead to unspecific reactions. The authors validate some of their findings using recombinant proteins, some of which being expressed in E coli, which may also be problematic due to different PTM patterns compared to human expression. The incubation of recombinant potential substrate with the enzyme and cosubstrate can again lead to artificial modifications that may not occur under physiological conditions and, unfortunately, I cannot see any kind of negative control for any of the three validation experiments, which is essential to ensure that not any protein will be modified in this setting.

2. In the same context, the authors used a method to "track phosphate release" which is not well-explained, stating that they preferred this over p-proteomics, as the phosphate release represents all p-sites, whereas MS will only show individual sites (and I guess may miss some based on sequence inaccessibility). Indeed, MS would be an important additional validation to ensure that there is not too many different p-sites on the recombinant proteins under the given conditions, as the identification of too many sites would support the note of low specificity in such experiments. Also the sites should be compared with Phosphositeplus.

3. Line 164: The authors speculate that secondary reactions would occur when using NADPH alone, as well, but this is not completely obvious to me --- do the authors refer to the same reactions?

4. Line 330: the authors discuss how the cutoffs for considering a protein as shifted can be changed and how this impacts the list of potential substrates. Using a suitable cutoff is of course a common problem in quantitative proteomics, but the authors have to discuss this in the context of the robustness of the method --- how is the variation in melting temperature between replicates? Indeed, the cutoff being used in the end should be based on this variation in order to produce more robust data, than by applying kind of arbitrary cutoffs.

5. Line 365-372. The authors speculate on the effects of PTMs on protein stability and that this will be hard to generalize. The beginning and end of this 8-line section is somewhat redundant. The authors should mention the concept of PTM crosstalk and the PTM code. Indeed it would be really

surprising if there was a general trend, as it seems that the entirety of PTMs dictate a proteins structure and function rather than individual PTMs, which may be one reason why the authors fail to verify some known substrates, they may simply be to heterogeneous within the sample which may even have functional/biological implications.

6. In general the writing has to be revised, as many statements throughout the manuscript are somewhat confusing or not ideal.

a. For instance the section between lines 98 and 101 should be rephrased as it is hard to read.

b. Lines 148-149: What do the authors mean with top 100-400 SIESTA proteins, this is not very specific.

c. What exactly is meant with “variable influence on projection” VIP values? It would be helpful for the reader to get a short statement on that, as it may not be common knowledge.

d. Lines 178-183: Please rephrase this part, it is confusing.

e. Line 232: Please summarize the assay to track phosphatase release in one sentence.

f. Line 252: Please mention that NAD is a cosubstrate, this is not obvious from the text.

g. Line 270 (and elsewhere): The term targeted MS is used in a wrong way. This typically refers to SRM/MRM and PRM assays, but what the authors used here is a triggered second MS/MS scan based on a decision-tree. Also the second MS/MS used EThcD (the term is not used though) while the corresponding figure 4c reads “targeted ETD MS/MS”, which is not correct in two ways.

h. Line 288: “coverage with trypsin digestion was not complete” – it was also not complete for the previously mentioned examples, and a 100% sequence coverage might be extremely rare.

i. line 307: although the hot spot TPP was briefly mentioned in the introduction, it would be helpful to have a short statement how it actually works, particularly as the authors refer to this repeatedly, e.g. line 318 where the statement is hard to access without actually knowing how it works.

j. line 402: I don't think a protein concentration can be distributed in aliquots.

k. line 343 (and was also somewhere else): please exchange ‘protein molecules’ which could refer to different proteins by ‘copies of the same protein’ or similar to clarify what is meant here.

Minor

1. Line 55: the authors should add work from the Krogan lab (PMID 22817900) to their introduction.

2. Line 59: Please add some references to the statement on using modified substrates as readout.

3. From the main text (e.g. figure 2) it is not obvious that Cys-levels were indeed checked using iodo-TMT. This was confusing and only made sense when reading the method section later.

4. The terms substrate and co-substrate in connection to enzymes are widely used. The authors should rephrase their statement on page 4, lines 88-90, which sounds as if this was specific for the manuscript at hand.

5. Line 322: The authors mention that different PTMs induce different thermal shifts. Please summarize the average/median shifts observed in the 3 different examples given in this study.

6. Line 347: The authors mention sub-cellular fractionation as a way to increase the depth of SIESTA. This implies that the lysis must be so mild, that organelles are still intact. The authors should comment on strengths and limitations of this.

7. The supplemental tables should have more explanations on what is actually presented. It would be good to have a better connection between the “raw” data and the lists of substrates, maybe by combining them as different sheets into a single table rather than having so many individual tables per experiment.

Reviewer #3 (Remarks to the Author):

“System-wide identification and prioritization of enzyme substrate by thermal analysis” submitted for publication in Nature Communication.

In this manuscript Amir Ata Saei et al. propose a workflow based on the mass spectrometry Cellular Thermal Shift Assay (MS-CETSA) to discover post-translational modifications produced by enzymes. Their hypothesis is that enzyme-induced post-translational modifications of substrate proteins may change the substrate thermal stability. This assumption is supported by multiple kinase-substrate pairs studied in vitro. Very recently this principle was generalized on a proteome-wide scale, as numerous shifts in protein melting temperatures were observed in response to site-specific phosphorylation events. Building on this concept, the authors aimed to detect post-translational modifications produced by, in principle, any enzymatic activity on its substrates with MS-CETSA. They found candidate protein substrates of three enzymes (TXNRD oxidoreductase, AKT1 kinase and poly-ADP ribose polymerase) by incubating human cell lysates with the enzymes purified in vitro and their co-factors. Their method named SIESTA is an interesting idea that could fill a gap in the field, as the (de)-stabilization events measured in MS-CETSA experiments are often difficult to interpret mechanistically. Although the screening of all potential substrates comes at the cost of producing recombinant enzymes, SIESTA would be very useful to reveal new enzyme-substrate associations. The method described in this manuscript could have an important impact in an area of biology where large scale studies are still scarce. However, I would suggest some conceptual and methodological clarifications of the content of this manuscript.

MAJOR POINTS:

CONCEPTUAL

1) The authors stated that the biological relevance of a protein modification correlates to the extent of the T_m shift that is observed in the thermal proteome profiling, and use this principle to prioritize substrates accordingly to their altered stability in their SIESTA assays. This idea follows the phosphorylation sites 'hotspots' theory proposed by Huang et al in a recent manuscript cited by the authors. However, two independent research groups questioned the conclusions of this work arguing that the experimental design was flawed. By reanalyzing the published data and performing independent experiments they showed that the extent of stability-altering phosphorylation is much less prevalent than what Huang et al proposed originally. At present, considering the open debate, I would consider the 'hotspot' theory at least controversial, therefore I would recommend to minimize references to the concepts originating from this paper and mitigate the statements regarding 'biological importance' and its association with 'larger T_m shifts'.

2) The criteria used for the definition of an enzyme substrate hit in SIESTA are not well described. What are the thresholds applied? How does the ranking of the hits work?

I understand that the specificity of the measured responses is calculated from the orthogonal partial least-square discriminant analysis (OPLS-DA) method referenced in the text. However, the authors should guide the readers summarizing what OPLS-DA does in a couple of sentences in the main text, as the audience will likely not know OPLS-DA.

The discrimination of the candidate hits seems to be based on the so called 'VIP value'. This concept is unclear for people who have not done an OPLS-DA analysis but it is the key

discriminant for the discovery of the highest stabilized proteins. This should be made more accessible for a non-specialist audience. For instance, with the current format it is not trivial to understand how to interpret the data shown in the figures. For instance, what do the acronyms 'pq[1]' and 'poso[1]' mean in the axes of the plot of figures 2d, S2c and S3c? What is a "negative reference context"? Since these three plots are showing the key results of the paper for the three enzymes under analysis, it is essential that sufficient elements are provided for the evaluation of this data.

3) There is a reference to the interaction databases GeneCards in line 147 of the main text and it is mentioned that GeneCard scores were also used to select putative substrates for validation. How the combination of the VIP and GeneCard scores were objectively used to prioritize or/and rank the candidate substrate?

4) A possible internal quality control for SIESTA could be to use the experiments with vehicle and co-substrate only. The Nordlund group and others have essentially done the same experiments before in these publications (<http://dx.plos.org/10.1371/journal.pone.0208273>, <http://www.nature.com/articles/s41467-019-09107-y>). What is the overlap among the ATP, NADPH and NAD datasets?

5) It is meaningfully pointed out that MS-CETSA experiments with vehicle and co-substrate find co-substrate binding proteins, as it has been extensively shown before. Can the authors comment whether they also observe significant T_m shifts between vehicle treated and enzyme treated samples? Is there any case in which substrate detection was achievable in absence of the co-substrate by adding recombinant enzymes to the lysates, as they do? Was any interaction between substrate and enzyme in absence of a post-translational modification observed in this dataset?

6) The duplicates of the 'vehicle only' melting curves figures 2e, 3b, 4b and in the supplementary figures should also be reported for completeness. They would provide a better perspective of how the different components of the binding reaction affect the substrate stability.

7) Adding the enzyme of interest in large quantities to protein lysates could potentially lead to artifact post-translational modification of proteins that are not substrates of the enzymes in physiological conditions. This important issue is not discussed. Was this considered at all when designing the experiments?

8) The text is at times hard to follow due to excessive use of abbreviations (ex lines 166-183).

This is also dangerous because it leads to misinterpretation. For instance, line 200, the authors write that "ACTB and MAP2K4 are new ATP binding proteins". I disagree. ACTB and MAPK4 are not novel ATP binding proteins. ACTB is cytoplasmic actin, and it is well known that ATP participates in the polymerization cycle of actin filaments. MAPK4 is a kinase component of the MAP kinase signal transduction pathway, and being a kinase binds ATP by definition.

9) Expressions like: "one could estimate the false positive rate to be not higher than 30%" (line 178) should be avoided if the sample size is 7. Similar issue at lines 300-301.

The discussion about "positive and -negative rates" should be toned down, if clear and objective criteria to define substrate hits are not provided, as discussed above (point 2 and 3).

EXPERIMENTAL:

10) The authors used separate multiplexed TMT10 samples to process and analyze experimental replicates in SIESTA experiments. Since this choice could introduce a certain experimental bias, can the authors show if this was taken into account for instance by examining the consistency between experimental duplicates?

Can the authors also explain why the AKT1 experiment require much more extensive peptide fractionation than the TXNRDq and PARP10 experiments (24 final fractions instead of 8)?

11) The validation of the monomer - dimer transition of GULP1 of supplementary figure 1d lacks a control. The total amount of GULP1 loaded in the two conditions should be shown in a SDS- PAGE gel run in denaturing conditions.

12) It would be interesting to comment about advantages/disadvantages of SIESTA versus REDOX proteomics for analyzing oxidoreductases substrates. The corresponding author's lab has a record of publications in the field of REDOX proteomics, so should be in a good position to briefly revise this in the discussion.

MINOR POINTS:

13) The scatter plot of Figure 2a does not show stabilization of known NADPH interacting proteins, rather consistency between replicates.

14) Some sentences are too vague. For instance, lines 135-138: "The analysis of specific ΔT_m shifts in the TXNRD1+NADPH treatment revealed that in the presence of NADPH, TXNRD1 destabilized both known and novel candidate substrate proteins (Supplementary Data 3). In general, the expected asymmetry in T_m shifts in favor of 138 destabilization was well pronounced (Fig. 2c)." Please report the exact number of destabilized and destabilized proteins in the main text to illustrate the content of figure 2c in the main text.

15) Lines 270-274: References to figure 4c and figure S3D appear to be missing.

Point-by-point response to the reviewers' comments

REVIEWER COMMENTS

Wherever the reviewer raised more than one issues in the same comment, we introduce [A..Z] marks and use them to trace the specific answers to each specific comment.

Reviewer #1 (Remarks to the Author):

In this work, Ata Saei and colleagues present a new approach to detect enzyme substrates based on thermal proteome profiling. The authors apply this approach to 3 enzymes, recovering some of their known substrates and identifying larger numbers of putative substrates. For each of those, the authors validate a small proportion that they prioritize, showing that the approach is generally valid – with a false positive rate ~20-30%, which I would deem acceptable for a proteome-wide method. The study is generally well designed and clearly written. I only have one major concern with the statistics used throughout the manuscript to identify potential substrates. I also have a couple of suggestions that I feel could make the approach more broadly applicable, and make it clearer for the readership what are the major limitations in its current form.

Response: We thank the reviewer for the positive appraisal of our work and the valuable comments.

Major comments

Comment 1. A. The authors used a t-test to calculate significance between conditions (together with a series of additional criteria properly described in the methods). However, it is not clear if the p-values from the t-test were corrected for multiple comparison. From browsing their code on github this does not seem to be the case. If not, this needs to be addressed, since the authors are performing thousands of tests for each experiment. **B.** As a suggestion, the authors might want to implement previous strategies, such as the ones in PMID:25278616 or PMID:31582558. The latter goes beyond T_m estimates and might increase the proteome space that the authors can probe (e.g., it could include TXN that the authors discuss does not properly melt).

Response: We thank the reviewer for this comment; indeed, choosing reliable hits with low rate of false positives is crucial for any proteomics methods.

A. Proper correction for multiple hypothesis requires an *a priori* knowledge of the error distribution, which was not available due to the complexity of our criteria for choosing SIESTA hits. Thus, to test our candidate selection procedure and determine the false discovery rate (FDR) among the SIESTA hits, in the pilot experiments, we performed permutation of the protein T_m values between the replicates within the dataset and applied to thus obtained nonsensical datasets the same selection criteria as for the unshuffled data. On average, FDR was 15%, which as the reviewer mentioned above, is acceptable for a proteomics method. Note that the top candidates with the largest ΔT_m values must have a much lower FDR. Our follow-up validation experiments also corroborate a similar FDR rate in practice. In conclusion, the cutoffs applied for identifying SIESTA hits are stringent and keep the rate of false discoveries to the minimum. A corresponding addition was made to the text.

B. In the first versions of this manuscript, we used the p-value strategies from the mentioned papers (PMID:25278616 or PMID:31582558), as we have done in our previous publication (PMID:32199331). With time, we realized that the above strategy leads to an increased level of false-positive and false-negative discoveries. The authors of the above publications must have also noticed these shortcomings and later recommended to manually inspect the T_m shifts after the analysis (PMID: 26379230). As manual inspection is notoriously subjective, we prefer to use a set of formal criteria as described in the text and as validated above. We also suggest using the more sophisticated OPLS-DA modeling scheme, for robust identification of enzyme substrates. Since OPLS-DA considers both significance and magnitude of an effect at once, selection of substrate candidates this way is statistically robust.

Comment 2. Related to the statistics, I do not fully understand why the authors use the PLS model to rank candidates and not simply the delta T_m (or the absolute delta T_m, or a proper significance value). From the methods section it is not clear what is the input for the PLS. Further, from for example Fig 2d the reference point for the single treatment is very close to proteins that show large negative shifts. Shouldn't these proteins be quite far from that reference point?

Response: PLS models consider both the magnitude of the input as well as its statistical significance, combining them in a single parameter (loading). The PLS input is the T_m for each protein under each

experimental condition and in each replicate. We found the PLS approach to be very suitable for SIESTA. The approach is now clearly explained upon first use in TXNRD1 results, with the explanation added to the Supplementary Figure 1. On Fig. 2d, the proteins that show a negative T_m shift in the combined enzyme-co-substrate treatment will be close to the single treatments reference point, while those with a positive shift will be close to the combination treatment reference point.

Comment 3. In all 3 examples, the authors mention the (known/putative) co-factor binding proteins, but never mention proteins that are (de)stabilized by the addition of the recombinant protein alone. By browsing the data for the TXNRD1 experiment, I found a few proteins that were strongly stabilized or destabilized by the protein alone (e.g., TRIO, CRADD) – I can imagine that the other two experiments have similar cases. Are these known interactors, or can the authors rationalize these in some way? Perhaps SIESTA could also be used to find new protein-protein interactions, making it even more exciting? It could also facilitate the expansion to heteromultimeric enzymes (see next comment), since the experiments could be performed first with enzyme vs control alone, find the partner enzymes and then use those together for the SIESTA experiment. I understand that this might be outside the scope of the current manuscript, but the authors should at least explore/discuss the results from enzyme vs control experiments.

Response: yes, SIESTA can also discover protein-protein interactions that manifest a shift in stability. Per reviewer request, we now include this information and discuss it for every system. For the AKT1 system, 4 proteins shifting in SIESTA were known from the literature to interact with this enzyme and one of them, MAZ, was known to be even modified with AKT1. For the PARP10 system, we performed a pull-down experiment and analyzed the overlap between PARP10 interactors and SIESTA results. There are 182 proteins shifting in response to PARP10 treatment. In pulldown experiment, we identified 115 proteins that are enriched more than log₂ fold change of >0.5. Of these, there are 6 proteins that show a shift in stability and enrichment in pulldown simultaneously. Fig. 5 and Supplementary Fig. 6 were added to cover these results.

Comment 4. Related to the last point, the authors should acknowledge in the discussion a few more drawbacks of SIESTA in its current form. First, it requires the availability of an active recombinant enzyme (i.e., the enzyme should not need to be activated in any way) that acts as a monomer or homomultimer. Second, the enzyme should only require a single co-factor that needs to be known and available.

Response: These are fair points; however, a non-active enzyme can also be used if it is activated to an equal degree in each SIESTA experiment. Thus we added to the manuscript the following: “One limitation with SIESTA is that the active (or equally activated in each experiment) recombinant enzyme must be used, and it should preferentially act as a monomer or homomultimer. Also, the co-substrate(s) need(s) to be known and available. On the other hand, SIESTA experimental design can be generalized to accommodate multiple (n) cofactors by introducing control experiments with (n-1) cofactors and the enzyme.”

Minor comments

Comment 1. While the criteria for calling hits in each experiment are explained in the methods section, it would be helpful that in each supplementary dataset there was a column included if a protein is a hit or not.

Response: This column is now added to the tables.

Comment 2. Line 148: “The average correlation between the VIP values of top 100-400 SIESTA proteins with their GeneCard scores (zero value was assigned if the protein was not present in the GeneCard list) was 0.18 (p<0.001).” This sentence is unclear, can the authors explain better what was done here?

Response: We deleted this analysis from the manuscript due to the collective comments from reviewers.

Comment 3. Line 174: “Interestingly, GULP1 exists as a dimer in vivo 32 and we noted the increased monomer levels for this protein upon incubation with TXNRD1+NADPH (Supplementary Fig. 1c-d).” Can this be caused by the slightly higher levels of TXNRD1 in the TXNRD1+NADPH experiment (by eye this seems to be higher in that experiment)?

Response: The same amount of TXNRD1 was loaded on the lanes. However, for the sake of clarity, we now report the ratio of the monomer to the monomer+dimer level.

Comment 4. Line 218: “31% (123/396) of the proteins annotated in Uniprot as ATP binders were also verified in our experiment.” The authors could also compare their results to previous TPP experiments with ATP, since even in lysate proteins that are stabilized are not necessarily binding ATP directly.

Response: Unfortunately, the processed data has not been made available in the previous studies. However, we present a comparison with UniProt, which has a compiled list of validated ATP, NADPH and NAD binding proteins. We have intentionally used 500 μM of ATP, at which concentration it mainly shows substrate activity, according to PMID: 30858367.

Comment 5. In Fig 2-4, when showing melting curves for potential substrates, I think the controls should also be included.

Response: We have now included the control melting curves for all the enzyme systems.

Andre Mateus

Reviewer #2 (Remarks to the Author):

Saei et al. describe an adaptation of the Thermal Proteome Profiling (TPP) methodology, they term SIESTA and which can be used to particularly screen for enzyme substrates. As the authors point out, proteome-wide screening for substrates of a particular enzyme is not necessarily straightforward.

The authors demonstrate the utility and versatility of SIESTA on three model enzymes, namely the thioredoxin reductase TXNRD1, the protein kinase AKT1, and poly-ADP-ribosyltransferase PARP-10 and show that known as well as novel substrates can be identified for these enzymes using a ‘differential’ approach. This is certainly an interesting adaptation of the TPP strategy that may address an existing need in screening for enzyme substrates, for instance when studying these enzymes as potential drug targets. Unfortunately, the presentation of the manuscript is far from optimal and a number of points listed below have to be addressed.

Response: We thank the reviewer for the positive appraisal of our work and the valuable comments.

Major

Comment 1. [A] The authors use a lysate (mild conditions) and add recombinant enzyme + substrate to study in vitro substrates of this enzyme. Although their approach undoubtedly generates interesting data, this strategy may be prone to false positive identifications. PTMs are mainly regulated spatially by bringing substrate and enzyme in close proximity (or not). Now, in a lysate this level of regulation is completely gone. I do not necessarily see a problem in using such a strategy, but I definitely miss a clear statement on this approach in order to make the reader aware of potential shortcomings. For instance, protein/substrate interactions may be specific to a certain organelle. [B] I was first also wondering about secondary effects, which the authors exclude based on the fact that the lysate is 77x fold diluted compared to the cell, but that also means that a huge excess of enzyme is added to drive the reaction which can lead to unspecific reactions. [C] The authors validate some of their findings using recombinant proteins, some of which being expressed in E coli, which may also be problematic due to different PTM patterns compared to human expression. [D] The incubation of recombinant potential substrate with the enzyme and cosubstrate can again lead to artificial modifications that may not occur under physiological conditions and, unfortunately, I cannot see any kind of negative control for any of the three validation experiments, which is essential to ensure that not any protein will be modified in this setting.

Response: We thank the reviewer for these critical comments. These issues are now mentioned in the discussion. [A, B] It reads “Furthermore, using lysate might distort the spatial regulation of enzyme substrate interaction and yield substrates that are not active in the biological context. The excess of the enzyme may also lead to unspecific reactions”. We tried to keep the conditions as physiological as possible. For example, in the case of ATP, we used 500 μM concentration where it mainly has cosubstrate activity (PMID: 30858367). The amount of added enzymes was not that high; we were actually limited by the amount of available recombinant enzymes. We now present the amount of added enzyme compared to

the total amount of enzyme found in untreated cell lysate. Within the current study, for the TXNRD1, AKT1 and PARP10 systems, the ratio of added enzyme was ~10, ~20 and ~1.5 fold compared to untreated lysate, respectively. Given the dilution of lysate by ≈ 77 fold, these ratios are well within the physiological concentrations. The Supplementary Figure 7 was added to illustrate the results. [C] We added “Moreover, recombinant proteins expressed in *E. coli* may lack some important PTMs necessary for their activity in human cells”. [D] We did not include an unrelated protein as a negative control during the incubation of recombinant potential substrate with the enzyme and cosubstrate (as it might create a bias). However, we have done other validation experiments on the same day and under the same exact conditions. For example, 7 redox proteomics experiments were performed in parallel for validating TXNRD1 substrates, with only 5/7 validated. For PARP10, we could not validate Caspase 6, meaning that our validation experiments did rule out false positives. For AKT1, we now complement our study with phosphoproteomics validation experiments where we use AKT1 inhibitors in cells. The results are presented in Fig. 3. In Figure 1 we included validation as a crucial step in the SIESTA workflow.

Comment 2. In the same context, the authors used a method to “track phosphate release” which is not well-explained, stating that they preferred this over p-proteomics, as the phosphate release represents all p-sites, whereas MS will only show individual sites (and I guess may miss some based on sequence inaccessibility). Indeed, MS would be an important additional validation to ensure that there is not too many different p-sites on the recombinant proteins under the given conditions, as the identification of too many sites would support the note of low specificity in such experiments. Also the sites should be compared with Phosphositeplus.

Response: In response to the reviewer comment, we performed phosphoproteomics experiments with AKT1 inhibitors added to living cells and now provide in several cases the potential sites of modifications changing protein stability. The validated proteins included some of the genuine substrates of AKT1 such as BCL3, TRIP12 and MEF2D, confirming our findings. This statement was also added: “For example, in two direct microarray screenings of human cells, 165 and 51 AKT1 substrates have been found, respectively, with no overlap between them. In the PhosphoSitePlus database containing an accumulated list of 206 human protein substrates for AKT1, there are 4 and 7 overlapping proteins with the two mentioned studies, respectively, while the 72 SIESTA substrates obtained with a 0.5°C cutoff, similarly gave 4 overlaps.”

Comment 3. Line 164: The authors speculate that secondary reactions would occur when using NADPH alone, as well, but this is not completely obvious to me --- do the authors refer to the same reactions?

Response: For better clarity, we rephrased this sentence that now reads: “Furthermore, if secondary reactions were present, they would also occur in lysates treated with NADPH alone (as the basal levels of cellular TXNRD1 was also present there), and thus would be filtered away in our analysis”.

Comment 4. Line 330: the authors discuss how the cutoffs for considering a protein as shifted can be changed and how this impacts the list of potential substrates. Using a suitable cutoff is of course a common problem in quantitative proteomics, but the authors have to discuss this in the context of the robustness of the method --- how is the variation in melting temperature between replicates? Indeed, the cutoff being used in the end should be based on this variation in order to produce more robust data, than by applying kind of arbitrary cutoffs.

Response: In choosing the final cutoff, we had also calculated the median variation between the replicates for different treatments for each enzyme system. This statement is now added: “The 1°C cutoff was chosen by analysis of the variation in T_m between replicates. In TXNRD1, AKT1 and PARP10 systems, the median T_m variation for different treatments was 0.50, 0.45 and 0.59 °C, respectively.” Furthermore, as now detailed in the text, our final cutoffs allow for a minimum false discovery rate, and are thus robust.

Comment 5. Line 365-372. The authors speculate on the effects of PTMs on protein stability and that this will be hard to generalize. The beginning and end of this 8-line section is somewhat redundant. The authors should mention the concept of PTM crosstalk and the PTM code. Indeed it would be really surprising if there was a general trend, as it seems that the entirety of PTMs dictate a proteins structure and function rather than individual PTMs, which may be one reason why the authors fail to verify some known substrates, they may simply be to heterogeneous within the sample which may even have functional/biological implications.

Response: This was a great comment and these arguments were added to the discussion section as “Given the complexity of the PTM cross-talk and the PTM code, the lack of an obvious general trend in ΔT_m shifts is not surprising, as the entirety of PTMs rather than individual PTM type dictate the structure and energetics of a given protein. This might also explain why we could not verify some known substrates, as it could be simply due to the heterogeneity of the PTM pool.”

Comment 6. In general the writing has to be revised, as many statements throughout the manuscript are somewhat confusing or not ideal.

a. For instance the section between lines 98 and 101 should be rephrased as it is hard to read.

Response: We revised and clarified the whole text. The specific section between lines 98 and 101 now reads: “Using orthogonal partial least squares-discriminant analysis (OPLS-DA), we create models where the proteins’ T_m values for the “enzyme + cosubstrate” treatment are contrasted against those of the other groups: “enzyme-treated” and “cosubstrate-treated” lysates.”.

b. Lines 148-149: What do the authors mean with top 100-400 SIESTA proteins, this is not very specific.

Response: We deleted this analysis from the manuscript due to the collective comments from reviewers.

c. What exactly is meant with “variable influence on projection” VIP values? It would be helpful for the reader to get a short statement on that, as it may not be common knowledge.

Response: VIP, OPLS-DA and other related parameters are now described in detail upon first use (the TXNRD1 results). And the Supplementary Figure 1 is added to clarify the modeling approach and interpretation of the results.

d. Lines 178-183: Please rephrase this part, it is confusing.

Response: This part was totally removed.

e. Line 232: Please summarize the assay to track phosphatase release in one sentence.

Response: This assay was summarized in one sentence.

f. Line 252: Please mention that NAD is a cosubstrate, this is not obvious from the text.

Response: This information was added.

g. Line 270 (and elsewhere): The term targeted MS is used in a wrong way. This typically refers to SRM/MRM and PRM assays, but what the authors used here is a triggered second MS/MS scan based on a decision-tree. Also the second MS/MS used EThcD (the term is not used though) while the corresponding figure 4c reads “targeted ETD MS/MS”, which is not correct in two ways.

Response: This was corrected accordingly.

h. Line 288: “coverage with trypsin digestion was not complete” – it was also not complete for the previously mentioned examples, and a 100% sequence coverage might be extremely rare.

Response: This statement was corrected and now reads: “For HDAC2 and PIN4, the sequence coverage with trypsin digestion was not sufficient to observe all potential modifications.”

i. line 307: although the hot spot TPP was briefly mentioned in the introduction, it would be helpful to have a

short statement how it actually works, particularly as the authors refer to this repeatedly, e.g. line 318 where the statement is hard to access without actually knowing how it works.

Response: We added the following clarification: “In this approach, after the thermal treatment of living cells and isolation of the soluble proteome, the lysate is divided in two aliquots for separate bulk proteome (5%) and phosphopeptide (95%) analyses, to uncover the link between the protein thermal stability and the phosphorylation state of that protein.”

j. line 402: I don't think a protein concentration can be distributed in aliquots.

Response: This sentence was corrected and now reads: “The protein concentration in the lysate was measured using Pierce BCA assay (Thermo) and the lysate was equally distributed into 8 aliquots (1 mL each).”

k. line 343 (and was also somewhere else): please exchange ‘protein molecules’ which could refer to different proteins by ‘copies of the same protein’ or similar to clarify what is meant here.

Response: “protein molecules” was changed to “protein copy numbers”.

Minor comments

Comment 1. Line 55: the authors should add work from the Krogan lab (PMID 22817900) to their introduction.

Response: Thanks for the suggestion, the citation was included.

Comment 2. Line 59: Please add some references to the statement on using modified substrates as readout.

Response: Two references were added.

Comment 3. From the main text (e.g. figure 2) it is not obvious that Cys-levels were indeed checked using iodo-TMT. This was confusing and only made sense when reading the method section later.

Response: The figure caption was corrected to clarify that iodoTMT labeling is used.

Comment 4. The terms substrate and co-substrate in connection to enzymes are widely used. The authors should rephrase their statement on page 4, lines 88-90, which sounds as if this was specific for the manuscript at hand.

Response: This statement was removed to satisfy this comment.

Comment 5. Line 322: The authors mention that different PTMs induce different thermal shifts. Please summarize the average/median shifts observed in the 3 different examples given in this study.

Response: The average absolute T_m shift was calculated for each system. This statement was added: “The average absolute shift for TXNRD1, AKT1 and PARP10 was 2.37-, 2.46- and 2.78 °C, respectively; and the median absolute shift for TXNRD1, AKT1 and PARP10 was 2.04, 2.10 and 2.19 °C, respectively.”

Comment 6. Line 347: The authors mention sub-cellular fractionation as a way to increase the depth of SIESTA. This implies that the lysis must be so mild, that organelles are still intact. The authors should comment on strengths and limitations of this.

Response: The following statement was added: “Such a strategy could probably identify organelle specific substrates; however, for that the lysis and fractionation should be mild enough to keep the organelle proteins intact.”

Comment 7. The supplemental tables should have more explanations on what is actually presented. It would be good to have a better connection between the “raw” data and the lists of substrates, maybe by combining them as different sheets into a single table rather than having so many individual tables per experiment.

Response: We merged the tables in the revised version of the manuscript.

Reviewer #3 (Remarks to the Author):

“System-wide identification and prioritization of enzyme substrate by thermal analysis” submitted for publication in Nature Communication.

In this manuscript Amir Ata Saei et al. propose a workflow based on the mass spectrometry Cellular Thermal Shift Assay (MS-CETSA) to discover post-translational modifications produced by enzymes. Their hypothesis is that enzyme-induced post-translational modifications of substrate proteins may change the substrate thermal stability. This assumption is supported by multiple kinase-substrate pairs studied in vitro. Very recently this principle was generalized on a proteome-wide scale, as numerous shifts in protein melting temperatures were observed in response to site-specific phosphorylation events. Building on this concept, the authors aimed to detect post-translational modifications produced by, in principle, any enzymatic activity on its substrates with MS-CETSA. They found candidate protein substrates of three enzymes (TXNRD oxidoreductase, AKT1 kinase and poly-ADP ribose polymerase) by incubating human cell lysates with the enzymes purified in vitro and their co-factors. Their method named SIESTA is an interesting idea that could fill a gap in the field, as the (de)-stabilization events measured in MS- CETSA experiments are often difficult to interpret mechanistically. Although the screening of all potential substrates comes at the cost of producing recombinant enzymes, SIESTA would be very useful to reveal new enzyme-substrate associations. The method described in this manuscript could have an important impact in an area of biology where large scale studies are still scarce. However, I would suggest some conceptual and methodological clarifications of the content of this manuscript.

Response: Thanks a lot for your nice comments on our manuscript.

MAJOR POINTS:

CONCEPTUAL

Comment 1. The authors stated that the biological relevance of a protein modification correlates to the extent of the T_m shift that is observed in the thermal proteome profiling, and use this principle to prioritize substrates accordingly to their altered stability in their SIESTA assays. This idea follows the phosphorylation sites ‘hotspots’ theory proposed by Huang et al in a recent manuscript cited by the authors. However, two independent research groups questioned the conclusions of this work arguing that the experimental design was flawed. By reanalyzing the published data and performing independent experiments they showed that the extent of stability-altering phosphorylation is much less prevalent than what Huang et al proposed originally. At present, considering the open debate, I would consider the ‘hotspot’ theory at least controversial, therefore I would recommend to minimize references to the concepts originating from this paper and mitigate the statements regarding ‘biological importance’ and its association with ‘larger T_m shifts’.

Response: Thanks for the comment. We also noticed the debates over the “hotspot” theory after we submitted the manuscript and have mitigated our citations to this paper in this version. Thus we added to the discussions: “In should be noted that recent findings from two independent groups raised doubts in the extent of stability-altering phosphorylation postulated by Huang et al.”. However, we still believe that substrates can be prioritized based on the size of the shift they exhibit in SIESTA.

Comment 2. The criteria used for the definition of an enzyme substrate hit in SIESTA are not well described. What are the thresholds applied? How does the ranking of the hits work?

Response: The selection criteria are now clearly stated in materials and methods: “For selection of putative substrates, the following criteria were used: 1) R² > 0.7 between the measurement and the fitted

curve, 2) the standard deviation between the replicates was $<2.5^{\circ}\text{C}$, 3) p values between the Enzyme-Cosubstrate treatment against Enzyme and Cosubstrate treatments < 0.05 for one condition and <0.1 for the other; 4) the absolute mean ΔT_m was larger than 1°C for both conditions (a similar approach was used for selection of cosubstrate binding proteins). We also now added: “The 1°C cutoff was chosen by analysis of the variation in T_m between replicates. In TXNRD1, AKT1 and PARP10 systems, the median T_m variation for different treatments was 0.50, 0.45 and 0.59°C , respectively.” We also added “The proteins passing the significance thresholds were ranked by absolute ΔT_m or VIP values obtained from OPLS-DA analysis”.

I understand that the specificity of the measured responses is calculated from the orthogonal partial least-square discriminant analysis (OPLS-DA) method referenced in the text. However, the authors should guide the readers summarizing what OPLS-DA does in a couple of sentences in the main text, as the audience will likely not know OPLS-DA.

The discrimination of the candidate hits seems to be based on the so called ‘VIP value’. This concept is unclear for people who have not done an OPLS-DA analysis but it is the key discriminant for the discovery of the highest stabilized proteins. This should be made more accessible for a non-specialist audience. For instance, with the current format it is not trivial to understand how to interpret the data shown in the figures. For instance, what do the acronyms ‘pq[1]’ and ‘poso[1]’ mean in the axes of the plot of figures 2d, S2c and S3c? What is a “negative reference context”? Since these three plots are showing the key results of the paper for the three enzymes under analysis, it is essential that sufficient elements are provided for the evaluation of this data.

Response: We have now described the OPLS-DA approach and the attributed parameters in the TXNRD1 results section (on the first use) and included the Supplementary Figure 1 to explain the approach and the way data must be interpreted. “pq[1]” and “poso[1]” were components 1 and 2 of the OPLS-DA model, and were changed accordingly in the new figures for clarity. The following paragraph was also added: “OPLS-DA is a multivariate supervised modeling tool for pinpointing the variables (here proteins) that have the largest discriminatory power between the two or more statistical groups (samples) 26. In the “loading plot” or “score scatter plot” for two-group comparison models, the predictive component is the x-axis, while y-axis is related to the orthogonal components that is irrelevant in this study (Supplementary Fig. 1a). In the loading plot (Supplementary Fig. 1b), each protein is represented by a dot. In SIESTA, the protein T_m for single treatments is contrasted with those from the combination treatment. The large dots on either side of the plot are the reference points for the treatments. Therefore, proteins specifically stabilized by the modification will move close to the reference point of the combination treatment and the destabilized proteins will be further away on the opposite side. The proximity of a protein to the reference point on either side of the x-axis is a measure of the magnitude of the thermal stability change upon modification and its reproducibility among the replicates. Each protein can also be characterized by the variable influence on projection (VIP-value). The VIP-values quantify the impact each variable (i.e., protein) has on the OPLS-DA model, with a higher value corresponding to a greater contribution. Thus the proteins with the highest VIP values are suitable as candidates for validation. For more detailed explanation, see Umetrics documentation.”

Comment 3. There is a reference to the interaction databases GeneCards in line 147 of the main text and it is mentioned that GeneCard scores were also used to select putative substrates for validation. How the combination of the VIP and GeneCard scores were objectively used to prioritize or/and rank the candidate substrate?

Response: We deleted this analysis from the manuscript due to the collective comments from reviewers.

Comment 4. A possible internal quality control for SIESTA could be to use the experiments with vehicle and co-substrate only. The Nordlund group and others have essentially done the same experiments before in these publications

(<http://dx.plos.org/10.1371/journal.pone.0208273>, <http://www.nature.com/articles/s41467-019-09107-y>). What is the overlap among the ATP, NADPH and NAD datasets?

Response: This is a valuable suggestion, but unfortunately, the processed data has not been made available in these studies. However, we present a comparison with UniProt, which has a compiled list of validated ATP, NADPH and NAD binding proteins.

Comment 5. It is meaningfully pointed out that MS-CETSA experiments with vehicle and co-substrate find co-substrate binding proteins, as is has been extensively shown before. [A] Can the authors comment whether they also observe significant T_m shifts between vehicle treated and enzyme treated samples? [B] Is there any case in which substrate detection was achievable in absence of the co-substrate by adding recombinant enzymes to the lysates, as they do? [C] Was any interaction between substrate and enzyme in absence of a post-translational modification observed in this dataset?

Response: [A] Yes, we have. Please see our response to the Comment 3 Reviewer 1. [B] Yes, there is. For instance, four proteins were known to interact with AKT1 in the literature. We performed a pulldown experiment using PARP10 and analyzed the overlap in PARP10 interactors with SIESTA results. These results are now fully described in the paper and Fig. 5 and supplementary Fig. 6 are added. Some of the identified hits were indeed known substrates of the enzymes under study, for example MAZ for AKT1 system. [C] Yes, there might be a few known substrates such as MAZ that interact with an enzyme, but do not show the additional stability change upon modification, but we have noticed that the reverse is more likely; i.e. a protein identified as a substrate be known as an interacting protein. The latter is discussed for AKT1 system in the text.

Comment 6. The duplicates of the ‘vehicle only’ melting curves figures 2e, 3b, 4b and in the supplementary figures should also be reported for completeness. They would provide a better prospective of how the different components of the binding reaction affect the substrate stability.

Response: We have now included the control melting curves for all the enzyme systems in all the figures.

Comment 7. Adding the enzyme of interest in large quantities to protein lysates could potentially lead to artifact post-translational modification of proteins that are not substrates of the enzymes in physiological conditions. This important issue is not discussed. Was this considered at all when designing the experiments?

Response: We thank the reviewer for the comment. These limitations of SIESTA are now discussed in detail in the discussion. Furthermore, we add “Since the addition of enzyme in excess can cause non-physiological modifications, the identified candidates should be validated by other techniques, as suggested in Fig. 1.” Yes, we tried to keep the conditions as physiological as possible. For example, in the case of ATP, we used 500 μM concentration where it mainly has cosubstrate activity (PMID: 30858367). The amount of added enzymes was not that high; we were actually limited by the amount of available recombinant enzymes. We now present the amount of added enzyme compared to the total amount of enzyme found in untreated cell lysate. Within the current study, for the TXNRD1, AKT1 and PARP10 systems, the ratio of added enzyme was ~ 10 , ~ 20 and ~ 1.5 fold compared to untreated lysate, respectively. Given the dilution of lysate by ≈ 77 fold, these ratios are well within the physiological concentrations. The Supplementary Figure 7 was added to illustrate the results.

Comment 8. The text is at times hard to follow due to excessive use of abbreviations (ex lines 166-183).

This is also dangerous because it leads to misinterpretation. For instance, line 200, the authors write that “ACTB and MAP2K4 are new ATP binding proteins”. I disagree. ACTB and MAPK4 are not novel ATP binding proteins. ACTB is cytoplasmic actin, and it is well known that ATP participates in the polymerization cycle of actin filaments. MAPK4 is a kinase component of the MAP kinase signal transduction pathway, and being a kinase binds ATP by definition.

Response: We reduced the number of abbreviations to increase clarity. Yes, ACTB and MAPK4 are known ATP interactors. The confusion is now fixed as “The melting curves of known ATP interactors ACTB and MAP2K4 are shown in Supplementary Fig. 3b.”

Comment 9. Expressions like: “one could estimate the false positive rate to be not higher than 30%” (line 178) should be avoided if the sample size is 7. Similar issue at lines 300-301.

The discussion about “positive and -negative rates” should be toned down, if clear and objective criteria to define substrate hits are not provided, as discussed above (point 2 and 3).

Response: These two statements were removed.

EXPERIMENTAL:

Comment 10. [A] The authors used separate multiplexed TMT10 samples to process and analyze experimental replicates in SIESTA experiments. Since this choice could introduce a certain experimental bias, can the authors show if this was taken into account for instance by examining the consistency between experimental duplicates?

[B] Can the authors also explain why the AKT1 experiment require much more extensive peptide fractionation than the TXNRD1 and PARP10 experiments (24 final fractions instead of 8)?

Response: [A] The consistency of the experimental replicates was checked by calculating the median variation in T_m between the replicates for different treatments for each enzyme system. “In TXNRD1, AKT1 and PARP10 systems, the median T_m variation for different treatments was 0.50, 0.45 and 0.59 °C, respectively.” This statement was now added to the paper.

[B] After successful proof-of-principle experiments on the TXNRD1 system with 8 fractions, we decided to investigate if higher number of fractions would improve the proteome coverage. The use of 24 fractions instead of 8 did for AKT1 increased the proteome coverage, but the magnitude of the increase did not justify in our opinion tripling the instrumental time. Therefore, we proceeded to analyze 8 fractions for the PARP10 system.

Comment 11. The validation of the monomer - dimer transition of GULP1 of supplementary figure 1d lacks a control. The total amount of GULP1 loaded in the two conditions should be shown in a SDS- PAGE gel run in denaturing conditions.

Response: This experiment with GULP1 and TXNRD1 was run in non-denaturing conditions. We now present the ratio of the monomer to dimer to avoid the issue of the total amount of the protein loaded.

Comment 12. It would be interesting to comment about advantages/disadvantages of SIESTA versus REDOX proteomics for analyzing oxidoreductases substrates. The corresponding author’s lab has a record of publications in the field of REDOX proteomics, so should be in a good position to briefly revise this in the discussion.

Response: We added a short paragraph to discussion: “Especially in the case of oxidoreductases, SIESTA may prove very useful. Although redox proteomics can also be used to analyze oxidoreductase substrates, it is impossible to cover all cysteine sites in all detected proteins. In contrast, the SIESTA information is obtained at the whole protein level, which by definition encompasses the whole sequence. This said, redox proteomics could still be an effective tool in verification of the oxidoreductase substrates.”

MINOR POINTS:

Comment 13. The scatter plot of Figure 2a does not show stabilization of known NADPH interacting proteins, rather consistency between replicates.

Response: We converted this plot to a volcano plot to fulfill the reviewer comments. The previous panel was moved to Supplementary Figure 2 to show the reproducibility between the replicates.

Comment 14. Some sentences are too vague. For instance, lines 135-138: “The analysis of specific ΔT_m shifts in the TXNRD1+NADPH treatment revealed that in the presence of NADPH, TXNRD1 destabilized both known and novel candidate substrate proteins (Supplementary Data 3). In general, the expected asymmetry in T_m shifts

in favor of 138 destabilization was well pronounced (Fig. 2c).” Please report the exact number of destabilized and stabilized proteins in the main text to illustrate the content of figure 2c in the main text.

Response: These sentences were clarified. The exact numbers of destabilized and stabilized proteins are now mentioned: “In general, as expected, there was an asymmetry in T_m shifts in favor of destabilization (63 destabilized proteins vs. 15 stabilized)”.

Comment 15. Lines 270-274: References to figure 4c and figure S3D appear to be missing.

Response: These figures were referenced later on that page, lines 283 and 287.

REVIEWERS' COMMENTS

Reviewer #1 (Remarks to the Author):

The authors have carefully addressed all of my previous comments, as well as those raised by the other two reviewers – the comments from the 3 reviewers largely overlapped. The manuscript is now even clearer, and I believe that this approach could make a significant impact in the discovery of enzyme substrates.

Reviewer #2 (Remarks to the Author):

The authors did an excellent job and have addressed all my comments.

Overall this is an exciting manuscript that has been further improved in the revision process. I recommend publication as is.

Rene Zahedi

Reviewer #3 (Remarks to the Author):

In general, my previous comments have been addressed, especially the important issues that, I noticed, were also raised by the other reviewers (general FDR control, better explanation of the data analysis procedure, possibility of protein-protein interactions detection). I'd just add a couple of remaining points that became apparent to me after this second read:

- in the introduction the authors should also mention applications based on stability of proteins from rates of oxidation (SPROX: PMID: PMID: 26530046) and limited proteolysis (LiP-MS: PMID: 29307493) to detect metabolite-protein interactions (so also substrate-enzyme relationship) on a proteome-wide scale.
- Regarding my previous comment 4: The author might have missed that the ATP protein binding data of the reference PMID: 30858367 is actually available as supplementary data 3.

- As for the figure caption of figure 5, I would change the title from 'TPP identifies protein-protein interactions' to 'TPP identifies protein interaction partners of enzymes added to cell lysates'. This describes more specifically the experiment that has been done there.

Point-by-point response to the reviewers' comments

REVIEWER COMMENTS

Reviewer #3 (Remarks to the Author):

Reviewer #3 (Remarks to the Author):

In general, my previous comments have been addressed, especially the important issues that, I noticed, were also raised by the other reviewers (general FDR control, better explanation of the data analysis procedure, possibility of protein-protein interactions detection).

Response: We thank the reviewer for all the valuable comments that helped increase the quality of the manuscript.

I'd just add a couple of remaining points that became apparent to me after this second read:

- in the introduction the authors should also mention applications based on stability of proteins from rates of oxidation (SPROX: PMID: 26530046) and limited proteolysis (LiP-MS: PMID: 29307493) to detect metabolite-protein interactions (so also substrate-enzyme relationship) on a proteome-wide scale.

Response: These two citations were added to the manuscript by adding this sentence: "Similar techniques such as stability of proteins from rates of oxidation (SPROX) and limited proteolysis can also be applied to detect metabolite-protein interactions 21,22."

- Regarding my previous comment 4: The author might have missed that the ATP protein binding data of the reference PMID: 30858367 is actually available as supplementary data 3.

Response: We analyzed the data and added the current statement: "Out of the 380 ATP-binding proteins identified in our experiment, 123 were annotated in UniProt as ATP binders and 161 were also identified in a two-dimensional TPP experiment on ATP in Jurkat cells".

- As for the figure caption of figure 5, I would change the title from 'TPP identifies protein-protein interactions' to 'TPP identifies protein interaction partners of enzymes added to cell lysates'. This describes more specifically the experiment that has been done there.

Response: The caption was edited according to reviewer comment.